# Discovery of 12O—A Novel Oral Multi-Kinase Inhibitor for the Treatment of Solid Tumor

**DOI:** 10.3390/molecules25215199

**Published:** 2020-11-09

**Authors:** Yan Fan, Zhi Huang, Xiaoshuang Wang, Yakun Ma, Yongtao Li, Shengyong Yang, Yi Shi

**Affiliations:** 1Department of Medicinal Chemistry, School of Medicine, Nankai University, 94 Weijin Road, Tianjin 300071, China; yanfan@nankai.edu.cn (Y.F.); huangzhi@mail.nankai.edu.cn (Z.H.); xswang@mail.nankai.edu.cn (X.W.); yakunma@nankai.edu.cn (Y.M.); liyt06198903@163.com (Y.L.); 2Medical Oncology, Cancer Center, State Key Laboratory of Biotherapy, West China Hospital, Sichuan University, Chengdu 610064, China; yangsy@scu.edu.cn

**Keywords:** CDK, FLT, inhibitor, cancer

## Abstract

A novel series of pyrimidine-benzotriazole derivatives have been synthesized and evaluated for their anticancer activity against human solid tumor cell lines. The most promising molecule **12O** was identified for its excellent antiproliferative activities, especially against the SiHa cell line with IC_50_ value as 0.009 μM. Kinase inhibition assay assessed **12O** was a potential multi-kinase inhibitor, which possessed potent inhibitory activities against cyclin-dependent kinases (CDKs) and fms-like tyrosine kinase (FLT) with IC_50_ values in the nanomolar range. Molecular docking studies illustrated that the introduction of triazole moiety in **12O** was critical for CDKs inhibition. In addition, **12O** inhibited cancer cell proliferation, colony-formation, and cell cycle progression and provoked apoptotic death in vitro. In an SiHa xenograft mouse model, a once-daily dose of compound **12O** at 20 mg/kg significantly suppressed the tumor growth without obvious toxicity. Taken together, **12O** provided valuable guide for further structural optimization for CDKs and FLT inhibitors.

## 1. Introduction

Cancer is an intricate and heterogeneous disease comprised of multistep and multigenetic processes. Many important genes aberrations severely disrupt the normal homoeostasis of cell growth and death [1]. Kinase inhibitors have been widely employed as molecularly targeted therapy drugs in clinical applications [2,3,4]. Individual molecular targets are often found to be less effective for treating complex disease, such as cancer, meaning that many single-target drugs cannot fully correct these diseases [5,6]. Multi-kinase inhibition is often a consequence of the simultaneous targeting of multiple pathways critical for tumor growth, including those contributing to proliferation, angiogenesis and apoptotic regulation. Compared with traditional agents, multi-kinase inhibitors that impact multiple targets simultaneously are better at controlling complex disease systems with superior efficacy, which are less prone to drug resistance, such as sorafenib, pazopanib, sunitinib, and dasatinib [7,8,9,10]. Therefore, rational drug discovery for more effective multi-target inhibitors may be necessary for cancer patients’ therapy.

Among the variety of molecular targets for cancer therapy, we are particularly interested in cyclin-dependent kinases (CDKs). CDKs have important roles in several cellular processes that regulate the cell division, apoptosis, transcription, and differentiation. The subtypes 1–4 and 6 of CDKs mainly mediate the cell cycle [11]. The CDK5 is associated with neuronal development and also plays a regulatory role in regulating various biological and pathological processes, including cancer progression [12]. The subtypes 7–9 of CDKs regulate transcription control [13]. Aberrations in CDKs and their partners have been observed in various tumors [14].

To date, multiple generations of CDK drugs have been designed and reported. The third-generation CDK drugs targeting CDK4/6, palbociclib[15], ribociclib[16] and abemaciclib[17] have received significant attention. However, in clinical trials, palbociclib and ribociclib needed to combine with letrozole or tamoxifen for the treatment of cancer, which indicated a lack of sensitivity to CDK4/6 monotherapy [18,19]. Abemaciclib had additional kinase activities, such as CDK1, CDK2, CDK9, GSK-3a/b, CaM-kinase II and so on, all of which may be beneficial for anticancer activity [20]. Simultaneous inhibition of CDKs has been shown to increase antitumor activities compared with the inhibition of a single CDK alone [21,22,23]. In this paper, we discovered a lead compound **12O** bearing 6-(pyrimidin-4-yl)-1H-benzo[d](1,2,3)triazole moiety as multi-target inhibitor against CDKs with IC_50_ values in the nanomolar range. Meanwhile, **12O** has an impact on several important targets, such as FLT. Inhibition of these targets is likely to contribute to the substantial antitumor activity. **12O** exhibited high potency against solid tumor, such as SiHa cells with an IC_50_ value of 0.009 μM. More importantly, **12O** exhibited excellent in vivo efficacy in SiHa xenograft model with low toxicity. This work implied that compound **12O** as a novel CDKs inhibitor deserved further research.

## 2. Results and Discussion

### 2.1. Chemistry

According to the literature [24], 1,2,3-triazole is one of the most fascinating nitrogen containing heterocycles, which has the ability to form effective non-covalent interactions with biological targets for discovery of potent anticancer agents. Many 1,2,3-triazole-containing compounds have immense importance in medicinal chemistry and exhibit anticancer activity. In order to discover novel active compounds to increase efficiencies against solid tumor, herein we introduced 1,2,3-triazole moiety and carried modifications based on **5J** of our in-house compound library (Appendix A) [25]. Meanwhile, according to the structural characteristic of CDK inhibitor, we introduced a CH_2_ group to synthesize **12A**-**P** (Appendix A).

The synthetic routes for all novel compounds are shown in Scheme 1 and Scheme 2. As shown in Scheme 1, commercial compound 4-bromo-2-fluoro-1-nitrobenzene (**1**) reacted with isopropylamine to give intermediate **2**, which underwent reduction reaction to provide compound **3**. Compound **3** was converted to **4** by closing the ring in the presence of conc. HCl and aq NaNO_2_. Then, **4** was heated with bis(pinacolato)diboronin to give **5**, which yielded **6**. As shown in Scheme 2, commercially available **7** reacted with aniline or substituted aniline to afford **8A**–**P**, which afforded **10A**–**P**. Then final compounds **12A**–**P** were generated by a palladium catalyzed cross-coupling reaction of compounds **10A**–**P** with compound **6**.

### 2.2. Biological Evaluation

#### 2.2.1. In Vitro Anticancer Screening

In vitro anticancer screening of all newly synthesized compounds towards human solid tumor cells, cervical (SiHa) and ovarian (SKOV3) cancer cell lines, was carried out by standard CCK-8 assay with cisplatin as the positive control, as cisplatin is one of the most used chemotherapy drugs. The half maximal inhibitory concentration (IC_50_) values of the tested compounds were listed in Table 1. Based on the observed results, most of the synthesized compounds exhibited good antiproliferative activity. Among them, molecule **12O** was found to be the most active compound in this series of compounds. Compound **12O** showed remarkable inhibitory effects against both cervical cancer cell line (SiHa) and ovarian cancer cell line (SKOV3) with IC_50_ values of 0.009 µM and 0.029 µM, respectively, showing more activity than the standard drug cisplatin. The compound was further evaluated in a dose response study against a panel of solid tumor cells, including cervical cancer cell lines (SiHa and HeLa), ovarian cancer cell lines (SKOV3 and OVCAR-5), breast cancer cell lines (4T1 and MCF-7), lung cancer cell lines (A549 and H460) to test cytotoxic activity. The IC_50_ values were summarized in Table 2, including the cytotoxic activity of the cisplatin used as reference compound. **12O** displayed obviously higher inhibition potency than positive control.

The most active compound **12O** with promising antiproliferative activities against the tested cell lines were subjected to further investigation for kinase inhibitory activities. Hit compound **5J** of our in-house compound library had a certain inhibitory activity against CDK4 (IC_50_: 0.046 μM) and VEGFR2 (vascular endothelial growth factor receptor, IC_50_: 0.046 μM) and showed moderate activity in SiHa cells (IC_50_: 0.050 μM). Kinase inhibition assays with a fixed concentration of 1 μM of compound **12O** were carried out against the CDKs and VEGFR family kinases. The results were displayed in Table 3. Further, the IC_50_ values of targets with inhibitory activities over 90% in preliminary enzymatic assays were detected. As shown, **12O** potently inhibited CDK2 (IC_50_ = 0.005 µM), CDK5 (IC_50_ = 0.006 µM), CDK9 (IC_50_ = 0.009 µM), and FLT4 (VEGFR3, IC_50_ = 0.068 µM). It has been reported that CDK2 is critically associated with tumor growth in multiple cancer types [26]. CDK5 plays important roles in regulating cancer progression [12]. Inhibition of CDK9 leads to the inhibition of proliferation and the promotion of apoptosis in many cancers by reducing many proteins up-regulated [27]. Previous studies have shown that FLT4 has been shown to promote lymphangiogenesis, angiogenesis, and proliferation in cervical, breast and other cancers [28,29]. Therefore, inhibition of FLT4 signaling may have a therapeutic benefit in limiting subsequent tumor cell dissemination [30]. The inhibitory activity of **12O** was also examined against PDGFR-related kinase because of their substantial sequence homology to the VEGFR family. **12O** showed selectivity versus the remaining PDGFR-related members and a range of unrelated tyrosine and serine/threonine kinases except FLT3 (IC_50_ = 0.024 µM) (Table 3). Most of the FLT3 inhibitors are applied for leukemia disease and some are used to treat solid tumors [22]. As reported, concurrent study of cell cycle arrest and angiogenesis inhibition was shown to be more effective due to synergy effects [31,32]. Thus, **12O** is a novel multi-target kinase inhibitor against CDKs and FLT, all of which are intimately associated with the growth, survival, and metastasis in tumor cells.

#### 2.2.2. Molecular Modeling of Compound **12O**

As indicated above, **12O** is a multi-target kinase inhibitor. Representative predicted binding modes of compound **12O** in CDKs were carried out. As reported, [26,33] ASP145, LEU83 and PHE82 were key residues within the ATP binding pocket among CDK2. As shown in Figure 1a, **12O** interacted with CDK2’s hinge region via the hydrogen of the NH of LEU83 residue. The triazole group exploited the hydrophobic region close to the residue PHE82 to form a favorable π-π interaction in CDK2. An additional hydrogen bond exists between the NH group of the compound **12O** and ASP145 residue. As shown in Figure 1b, **12O** tightly hydrogen-bonded with the kinase hinge regions in the ATP-binding pocket of CDK5. The triazole of **12O** formed two hydrogen bonds with the LYS128 residue in CDK5. In addition, LYS33, GLY146, and TRY15 possessed a key role in positioning and binding of **12O** through hydrogen bonds [34]. As shown in Figure 1c (PDB: 4BCF), an important hydrogen bond existed between the carbonyl group of the compound **12O** and LYS151 residue of CDK9. The triazole of **12O** forms a key hydrogen bond to the backbone ASP167 residue. Meanwhile, at the back of the ATP binding site, the triazole group exploited the hydrophobic region close to the gatekeeper residue PHE103 to form a favorable π-π interaction in CDK9. Therefore, from Figure 1a–c, we can see that the great flexibility of **12O** enabled CDKs to be effectively accommodated. The triazole substitute group was beneficial for the binding efficacy of **12O** with kinase protein.

Therefore, the respective predicted binding models explained the high potency of **12O** toward CDKs. Analysis of molecular docking models showed a good accommodation of **12O** onto the binding sites of CDKs, with the establishment of different contacts, such as hydrogen bonds, π-π interactions, with the key residues of the investigated proteins.

#### 2.2.3. **12O** Inhibited Clonogenicity In Vitro

Colony formation assay assesses the effect of a therapy on clonogenic survival which is one of the most important parameters of therapy efficacy in oncology. As the excellent inhibitory activity showed by **12O** over SiHa cell line, we selected SiHa as representative model to detect the antitumor activities of **12O** in vitro and in vivo. As shown in Figure 2, **12O** dose-dependently decreased the formation of colonies in all the SiHa cell lines. Colony formation assays confirmed that **12O** groups significantly reduced clonogenicity compared with the other treatments. We also performed colony formation assay in SKOV3 cells at concentrations according to IC_50_ values to confirm the result (Appendix A).

To understand how **12O** inhibited cell proliferation, cell cycle analysis was performed. The effect of **12O** on cell cycle distribution was assessed in the SiHa cell lines. The results indicated that **12O** dose-dependently blocked the cell cycle at G2/M phase compared to the vehicle-treated cells (Figure 3). Of note, **12O** at 0.04 μM was able to increase the proportion of the cells in the G2/M phase from approximately 16.0 ± 1.3% (DMSO) to 22.3 ± 0.7%, and decreased the corresponding fraction of the cells in the S phase from about 22.8 ± 0.4 % (DMSO) to 16.1 ± 1.1 % (Figure 3). Therefore, our results suggested that **12O** inhibited cell proliferation by arresting cells in G2/M phase of the cell cycle.

#### 2.2.4. **12O** Induced Cell Apoptosis In Vitro

Subsequently, the effect of **12O** on cell apoptosis was performed using Annexin V- FITV/PI staining. As shown in Figure 4, **12O** treatment induced apoptosis of SiHa cells in a concentration-dependent manner. Compared with the cisplatin and vehicle condition, **12O** increased the apoptotic cells (Q2 late apoptotic plus Q3 early apoptotic). This study illustrated that compound **12O** could induce cell death via apoptosis.

#### 2.2.5. In Vivo Anti-Tumor Activity of **12O**

Based on a desirable set of in vitro properties, **12O** was finally selected for further in vivo evaluation. We proceeded to test the antitumor efficacy of **12O** in SiHa xenograft mice. Animals were orally given three doses (5, 10, and 20 mg Kg^−1^) of **12O**, one dose (20 mg Kg^−1^) of cisplatin (positive control), or vehicle (control) every day. No obvious toxicity was observed in all the treated groups (Figure 5a). As shown in Figure 5, the growth of xenograft tumors was significantly inhibited by **12O** in a dose-dependent manner. Tumor growth inhibitions of 51.25%, 65.52%, 79.29% were observed at doses of 5, 10, and 20 mg/kg, respectively. In contrast, cisplatin (20 mg/kg) as the positive control was less potent (31.14%) compared with **12O** at the same dose. Of note, In Figure 5c, an obvious decrease in tumor size was also observed at the end of observation. The average tumor weight (Figure 5d) of the **12O**-treated group was less than that of cisplatin. HE staining was further performed to identify the efficacy of **12O** (Figure 5e). It was showed that the nuclei of tumor cells in vehicle controls were large and hyperchromatic, while the nuclei of **12O**-treated tumor cells were pyknotic. HE staining results in tumor tissues treated with **12O** further demonstrated the inhibition of tumor growth. Experiments in vivo demonstrated that **12O** did have significant antitumor activity compared with cisplatin, which is used as a first-line chemotherapy drug against solid tumor in clinical settings. **12O** as a novel multi-target kinase inhibitor effectively inhibited tumor growth of mice without obvious toxicity. Additional kinase activities of **12O** intimately associated with the growth, survival, and metastasis in tumor cells, may contribute to the antitumor activity, but may also cause side effects. Multi-target kinase inhibitor extensively used in clinic cancer therapy, while hampered by associated adverse reactions and side-effect. Thus, further evidence is needed. An in-depth study of compound **12O** is ongoing in our laboratory and will be reported in due course.

## 3. Materials and Methods 

^1^H NMR (400 MHz) and ^13^C NMR (101 MHz) spectra were taken on a Bruker AV-400 MHz spectrometer, Bruker corporation, Karlsruhe, Germany. High Performance Liquid Chromatography (HPLC) was manufactured by Shimadzu, Kyoto, Japan. High-resolution mass spectra (HRMS) was manufactured by VG Instruments Ltd., London, UK. Flow cytometer was manufactured by Becton Dickinson (BD), San Jose, USA. Microscope was manufactured by Olympus, Tokyo, Japan. Cell incubator was manufactured by Thermo Fisher Scientific, Inc., Waltham, MA, USA. The ultra-pure water was supplied by a Milli-Q water purification system manufactured by EMD Millipore Corporation (Bedford, MA, USA).

### 3.1. Chemistry

#### 3.1.1. General Information

The commercially obtained chemicals were used directly without further purification. Solvents were purified and distilled following the standard procedures. All the reactions were monitored by thin-layer chromatography (TLC). The NMR spectra were taken on a Bruker AV-400 MHz spectrometer (400 MHz for ^1^H and 101 MHz for ^13^C) and chemical shifts were expressed in ppm downfield using tetramethylsilane as the internal standard. HRMSwere performed on a VG ZAB-HS mass spectrometer under electron spray ionization (ESI). All the derivatives for testing bioactivity were purified to ≥ 95% purity which was determined by HPLC analysis on a Shimadzu Prominence-i LC-2030C 3D system (column, InertSustain C_18_, 4.6 mm × 250 mm, 5 μM; mobile phase, gradient elution of methanol/H_2_O (90:10); low rate, 1.0 mL/min; UV wavelength, 190−800 nm; temperature, 40 °C; injection volume, 10 μL).

#### 3.1.2. Synthesis of Intermediate Compound **6**

5-bromo-N-isopropyl-2-nitroaniline(**2**). An orange mixture of 4-bromo-2-fluoro-1 -nitrobenzene (4 g, 18.18 mmol), isopropylamine (1.7 mL, 20 mmol) and K_2_CO_3_ (2.51 g, 36.36 mmol) in DMF (40 mL) was stirred at room temperature overnight. The resulting mixture was diluted with water and the mixture was extracted by ethyl acetate, and the combined organic layers were washed by water and brine, and dried by anhydrous magnesium sulfate. The solvent was evaporated, and the residue was purified by silica gel column chromatography to obtain **2** (4.52 g, 96%) as a bright orange solid. ^1^H NMR (400 MHz, DMSO-*d_6_*) δ 8.08–7.77 (m, 2H), 7.20 (d, *J* = 2.5 Hz, 1H), 6.76 (dd, *J* = 9.1, 2.3 Hz, 1H), 4.08–3.77 (m, 1H), 1.23 (dd, *J* = 6.4, 2.3 Hz, 6H). ^13^C NMR (101 MHz, DMSO-*d_6_*) δ 145.12, 131.60, 130.47, 128.55, 118.48, 117.14, 44.09, 22.51.

*5-bromo-N^1^-isopropylbenzene-1,2-diamine(**3**).* Compound **2** (4 g, 15.4 mmol) was dissolved in AcOH (100 mL) and Fe powder (8.6 g, 154 mmol, 10 equivalents) was added. The mixture was stirred at RT for 0.5 h and then heated at 70 °C for 3 h. The suspension was basified with aqNaHCO_3_, diluted with CH_2_Cl_2_ (150 mL), and filtered through celite. The filtrate was extracted with CH_2_Cl_2_ (2 × 100 mL) and the combined organic phases were dried (K_2_CO_3_) and evaporated to furnish **3** (3.2 g, 91%) as a dark solid. The crude product **3** was not subjected to further purification and was taken directly.

*6-bromo-1-isopropyl-1H-benzo[d](1,2,3)triazole*(**4**). To a 0 °C mixture of the crude product **3** (3.2 g, 14 mmol) in conc. HCl (40 mL) was added NaNO_2_(1.1g,15.4 mmol) in H_2_O (10 mL). The mixture was allowed to warm to room temperature and was stirred for 1 h. After recooling to 0 °C, the mixture was treated with 6N NaOH until basic, the precipitate was then filtered, rinsed with H_2_O and dried to afford **4** (2.62 g, 78%). ^1^H NMR (400 MHz, DMSO-*d_6_*) δ 8.27 (d, *J* = 2.4 Hz, 1H), 7.98 (dd, *J* = 8.9, 2.3 Hz, 1H), 7.48 (dq, *J* = 8.8, 1.6 Hz, 1H), 5.21 (td, *J* = 6.7, 2.3 Hz, 1H), 1.59 (dd, *J* = 6.8, 2.4 Hz, 6H). ^13^C NMR (101 MHz, DMSO-*d_6_*) δ 147.32, 132.32, 129.26, 119.05, 117.79, 84.53, 83.27, 51.07, 25.27, 25.14, 22.75.

*1-isopropyl-6-(4,4,5,5-tetramethyl-1,3,2-dioxaborolan-2-yl)-1H benzo[d](1,2,3) triazole*(**5**). To a suspension of compound (**4**) (3 g, 12.5 mmol) in 50 mL 1,4-dioxane the following substances were added: bis(pinacolato)diboron (3.81 g, 15 mmol 1.2 equiv), PdCl_2_(pddf) (457 mg, 0.625 mmol, 0.05 equiv) and KOAc (4.3 g, 43.8 mmol, 3.5 equiv) and the flask was purged with N_2_. Then the flask was sealed and the mixture was heated for 12 h at 95℃. The reaction was cooled to RT, the solvent was removed under reduced pressure, and the residue was purified by silica gel column chromatography to obtain **5** (2.4 g, 67%).^1^H NMR (400 MHz, DMSO-*d_6_*) δ 8.17 (s, 1H), 8.01 (d, *J* = 8.3 Hz, 1H), 7.64 (d, *J* = 8.4 Hz, 1H), 5.33 (p, *J* = 6.7 Hz, 1H), 1.61 (d, *J* = 6.7 Hz, 6H), 1.32 (s, 12H).^13^C NMR (101 MHz, DMSO-*d_6_*) δ 147.32, 132.32, 129.26, 119.05, 117.79, 83.27, 51.07, 25.14, 22.75.

*6-(2-chloro-5-fluoropyrimidin-4-yl)-1-isopropyl-1H-benzo[d](1,2,3) triazole(**6**).* To a suspension of compound (**5**) (2 g, 7 mmol) in 50 mL 1,4-dioxane and 5 mL H_2_O, the following substances were added: 2,4-dichloro-5-fluoropyrimidine (2.3 g, 14 mmol, 2 equiv), PdCl_2_(pddf) (512 mg, 0.7 mmol, 0.1 equiv) and NaHCO_3_ (1.8 g, 21 mmol, 3 equiv) and the flask was purged with N_2_. Then the flask was sealed and the mixture was heated for 12 h at 100 °C. The reaction was cooled to RT, the solvent was removed under reduced pressure, and the residue was purified by silica gel column chromatography to obtain **6** (1.2 g, 59%).^1^H NMR (400 MHz, DMSO-*d_6_*) δ 8.98 (d, *J* = 3.0 Hz, 1H), 8.46 (s, 1H), 8.16 (d, *J* = 8.7 Hz, 1H), 8.03–7.82 (m, 1H), 5.34 (p, *J* = 6.7 Hz, 1H), 1.64 (d, *J* = 6.6 Hz, 6H). ^13^C NMR (101 MHz, DMSO-*d_6_*) δ 155.49 (d, *J* = 265.4 Hz), 154.70 (d, *J* = 3.6 Hz), 153.95 (d, *J* = 10.1 Hz), 150.48 (d, *J* = 26.8 Hz), 146.76, 132.36, 130.17 (d, *J* = 5.1 Hz), 124.57 (d, *J* = 5.8 Hz), 120.16, 112.80 (d, *J* = 7.3 Hz), 51.58, 22.57.

#### 3.1.3. General Procedure for the Preparation of 8A-P

To a stirred solution of cyclopropane-1,1-dicarboxylic acid (**7**) (13.01 g, 100 mmol) in isopropyl acetate (150 mL) at 0 °C, thionyl chloride (12.5 g, 105 mmol) was added dropwise. After addition, the resulting mixture was stirred at room temperature and stirred for 6 h. The resulting mixture was then treated with a solution of aniline or substituted aniline (110 mmol) and triethylamine (110 mmol) in isopropyl acetate (40 mL) over 1 h. After stirring for 2 h, the resulting mixture was added to ethyl acetate (500 mL). The solvent was washed by 1 N HCl solution and brine. The organic phase was dried over MgSO4, evaporated and the residue treated with heptane (200 mL). The product slurry was stirred for 0.5 h, filtered and dried under vacuum to obtain product.

*1-((3-fluorophenyl)carbamoyl)cyclopropane-1-carboxylic acid* (**8A**). White solid (76%), ^1^H NMR (400 MHz, DMSO-*d*_6_) δ 13.12 (s, 1H), 10.73 (s, 1H), 7.61 (dt, *J* = 11.3, 2.0 Hz, 1H), 7.41–7.27 (m, 2H), 6.88 (ddt, *J* = 9.1, 6.7, 2.7 Hz, 1H), 1.41 (s, 4H).

*1-((4-fluorophenyl)carbamoyl)cyclopropane-1-carboxylic acid* (**8B**). White solid (67%), ^1^H NMR (400 MHz, DMSO-*d*_6_) δ 13.11 (s, 1H), 10.81–10.50 (m, 1H), 7.63 (dd, *J* = 8.9, 5.0 Hz, 2H), 7.13 (tt, *J* = 8.9, 3.0 Hz, 2H), 1.54–1.30 (m, 4H).

*1-((3-chlorophenyl)carbamoyl)cyclopropane-1-carboxylic acid* (**8C**). White solid (81%), ^1^H NMR (400 MHz, DMSO-*d*_6_) δ 13.12 (s, 1H), 10.70 (s, 1H), 7.83 (t, *J* = 1.9 Hz, 1H), 7.46 (d, *J* = 8.2 Hz, 1H), 7.32 (td, *J* = 8.1, 1.5 Hz, 1H), 7.10 (dd, *J* = 7.9, 2.1 Hz, 1H), 1.42 (s, 4H); ^13^C NMR (101 MHz, DMSO-*d_6_*) δ 173.52, 167.03, 140.20, 133.04, 130.33, 123.03, 118.79, 117.69, 28.87, 16.98.

*1-((4-chlorophenyl)carbamoyl)cyclopropane-1-carboxylic acid (**8D**).* White solid (55%), ^1^H NMR (400 MHz, DMSO-*d*_6_) δ 13.10 (s, 1H), 10.65 (s, 1H), 7.71–7.54 (m, 2H), 7.49–7.23 (m, 2H), 1.41 (s, 4H); ^13^C NMR (101 MHz, DMSO-*d_6_*) δ 173.60, 166.80, 137.75, 128.56, 126.87, 120.85, 28.80, 16.95.

*1-((3-bromophenyl)carbamoyl)cyclopropane-1-carboxylic acid* (**8E**). White solid (76%), ^1^H NMR (400 MHz, DMSO-*d*_6_) δ 13.12 (s, 1H), 10.69 (s, 1H), 7.97 (t, *J* = 2.0 Hz, 1H), 7.50 (dt, *J* = 7.5, 1.9 Hz, 1H), 7.39–7.15 (m, 2H), 1.42 (s, 4H); ^13^C NMR (101 MHz, DMSO-*d_6_*) δ 174.03, 167.51, 140.83, 131.13, 126.43, 122.14, 122.01, 118.57, 29.38, 17.50.

*1-((4-bromophenyl)carbamoyl)cyclopropane-1-carboxylic acid* (**8F**). White solid (84%), ^1^H NMR (400 MHz, DMSO-*d*_6_) δ 13.13 (s, 1H), 10.70 (s, 1H), 7.64–7.55 (m, 2H), 7.51–7.41 (m, 2H), 1.43 (s, 4H); ^13^C NMR (101 MHz, DMSO-*d_6_*) δ 174.22, 167.32, 138.61, 131.95, 121.73, 115.43, 29.15, 17.70.

*1-((3-methoxyphenyl)carbamoyl)cyclopropane-1-carboxylic acid* (**8G**). White solid (65%), ^1^H NMR (400 MHz, DMSO-*d*_6_) δ 13.09 (s, 1H), 10.62 (s, 1H), 7.33 (t, *J* = 2.1 Hz, 1H), 7.20 (t, *J* = 8.1 Hz, 1H), 7.13 (dt, *J* = 8.2, 1.3 Hz, 1H), 6.63 (ddd, *J* = 8.1, 2.5, 1.0 Hz, 1H), 3.72 (s, 3H), 1.43 (s, 4H); ^13^C NMR (101 MHz, DMSO-*d_6_*) δ 174.40, 167.20, 159.99, 140.42, 129.95, 112.05, 109.40, 105.50, 55.40, 29.01, 17.77.

*1-((4-methoxyphenyl)carbamoyl)cyclopropane-1-carboxylic acid* (**8H**). Off-white solid (70%), ^1^H NMR (400 MHz, DMSO-*d*_6_) δ 10.49 (s, 1H), 7.60–7.31 (m, 2H), 7.12–6.68 (m, 2H), 3.71 (s, 3H), 1.42 (s, 4H); ^13^C NMR (101 MHz, DMSO-*d_6_*) δ 174.56, 166.77, 155.78, 132.36, 121.41, 114.27, 55.57, 28.59, 17.81.

*1-(m-tolylcarbamoyl)cyclopropane-1-carboxylic acid* (**8I**). White solid (75%), ^1^H NMR (400 MHz, DMSO-*d*_6_) δ 13.14 (s, 1H), 10.58 (s, 1H), 7.52–7.31 (m, 2H), 7.17 (t, *J* = 7.8 Hz, 1H), 6.87 (d, *J* = 7.4 Hz, 1H), 2.27 (s, 3H), 1.42 (s, 4H).

*1-(p-tolylcarbamoyl)cyclopropane-1-carboxylic acid* (**8J**). White solid (79%), ^1^H NMR (400 MHz, DMSO-*d*_6_) δ 13.11 (s, 1H), 10.54 (s, 1H), 7.60–7.36 (m, 2H), 7.20–6.99 (m, 2H), 2.25 (s, 3H), 1.42 (s, 4H).

*1-((3-cyanophenyl)carbamoyl)cyclopropane-1-carboxylic acid* (**8K**). White solid (79%), ^1^H NMR (400 MHz, DMSO-*d*_6_) δ 13.15 (s, 1H), 10.84 (s, 1H), 8.11 (q, *J* = 1.3 Hz, 1H), 7.84 (ddd, *J* = 6.0, 3.3, 2.2 Hz, 1H), 7.69 – 7.39 (m, 2H), 1.43 (s, 4H); ^13^C NMR (101 MHz, DMSO-*d_6_*) δ 173.88, 167.82, 140.06, 130.65, 127.37, 124.38, 122.49, 119.12, 112.03, 29.46, 17.44.

*1-((4-cyanophenyl)carbamoyl)cyclopropane-1-carboxylic acid* (**8L**). Yellow solid (79%), ^1^H NMR (400 MHz, DMSO-*d*_6_) δ 13.13 (s, 1H), 10.90 (s, 1H), 7.90–7.62 (m, 4H), 1.42 (s, 4H); ^13^C NMR (101 MHz, DMSO-*d_6_*) δ 173.76, 167.90, 143.48, 133.70, 119.78, 119.49, 105.53, 29.85, 17.29.

*1-((3-(trifluoromethyl)phenyl)carbamoyl)cyclopropane-1-carboxylic acid* (**8M**). White solid (81%), ^1^H NMR (400 MHz, DMSO-*d*_6_) δ 13.13 (s, 1H), 10.83 (s, 1H), 8.13 (t, *J* = 1.9 Hz, 1H), 7.95–7.72 (m, 1H), 7.53 (t, *J* = 8.0 Hz, 1H), 7.46–7.30 (m, 1H), 1.43 (s, 4H); ^13^C NMR (101 MHz, DMSO-*d_6_*) δ 173.92, 167.75, 140.04, 130.39, 130.34, 130.08, 129.77, 129.45, 128.60, 125.89, 123.33, 123.18, 120.13, 120.09, 115.89, 115.85, 29.49, 17.34.

*1-((4-(trifluoromethyl)phenyl)carbamoyl)cyclopropane-1-carboxylic acid* (**8N**). Yellow solid (82%), ^1^H NMR (400 MHz, DMSO-*d*_6_) δ 12.95 (s, 1H), 10.91 (s, 1H), 7.82 (d, *J* = 8.5 Hz, 2H), 7.65 (d, *J* = 8.5 Hz, 2H), 1.44 (s, 4H); ^13^C NMR (101 MHz, DMSO-*d_6_*) δ 174.04, 167.78, 142.80, 126.48, 126.45, 126.41, 126.13, 124.03, 123.71, 123.43, 119.65, 29.44, 17.55.

*1-(phenylcarbamoyl)cyclopropane-1-carboxylic acid* (**8O**). White solid (66%), ^1^H NMR (400 MHz, DMSO-*d*_6_) δ 13.10 (s, 1H), 10.60 (s, 1H), 7.63–7.54 (m, 2H), 7.40–7.27 (m, 2H), 7.11–6.99 (m, 1H), 1.42 (s, 4H).

*1-((4-chloro-3-(trifluoromethyl)phenyl)carbamoyl)cyclopropane-1-carboxylic acid* (**8P**). White solid (67%), ^1^H NMR (400 MHz, DMSO-*d*_6_) δ 13.13 (s, 1H), 10.88 (s, 1H), 8.20 (d, *J* = 2.3 Hz, 1H), 7.85 (d, *J* = 8.8 Hz, 1H), 7.71–7.43 (m, 1H), 1.42 (s, 4H); ^13^C NMR (101 MHz, DMSO-*d_6_*) δ 173.24, 167.32, 138.23, 131.95, 131.91, 126.82, 126.51, 124.01, 121.28, 117.97, 117.91, 29.02, 16.75.

#### 3.1.4. General Procedure for the Preparation of **10A−P**

To a two-necked flask, compound **8A–P** (30 mmol), 4-(aminomethyl)aniline dihydrochloride (30 mmol), EDCI (45 mmol), HOBt (36 mmol), DIEA (120 mmol) and DMF (30 mL) were charged. The mixture was stirred at room temperature for 12 h, then quenched by water. The mixture was extracted by ethyl acetate, and the combined organic layers were washed by saturated aqueous NaHCO_3_ solution, water and brine, dried by anhydrous magnesium sulfate. The solvent was evaporated, and the residue was purified by silica gel column chromatography.

*N-(4-(aminomethyl)phenyl)-N-(3-fluorophenyl)cyclopropane-1,1-dicarboxamide*(**10A**). Brown solid (69%), ^1^H NMR (400 MHz, DMSO-*d*_6_) δ 10.91 (s, 1H), 8.27 (s, 1H), 7.63 (d, *J* = 11.6 Hz, 1H), 7.32 (d, *J* = 4.4 Hz, 2H), 6.91 (dd, *J* = 21.6, 5.2 Hz, 3H), 6.51 (d, *J* = 8.1 Hz, 2H), 4.96 (s, 2H), 4.15 (d, *J* = 5.8 Hz, 2H), 1.39 (d, *J* = 3.1 Hz, 4H). ^13^C NMR (101 MHz, DMSO-*d*_6_) δ 170.18, 168.48, 162.05 (d, *J* = 240.9 Hz), 147.48, 140.40 (d, *J* = 11.0 Hz), 130.14 (d, *J* = 9.6 Hz), 128.15, 126.10, 115.54, 113.68, 109.85 (d, *J* = 20.9 Hz), 106.67 (d, *J* = 26.2 Hz), 42.35, 29.52, 15.92.

*N-(4-(aminomethyl)phenyl)-N-(4-fluorophenyl)cyclopropane-1,1-dicarboxamide* (**10B**). Brown solid (78%),^1^H NMR (400 MHz, DMSO-*d*_6_) δ 10.69 (s, 1H), 8.31 (d, *J* = 5.9 Hz, 1H), 7.61 (dd, *J* = 8.8, 5.0 Hz, 2H), 7.13 (t, *J* = 8.7 Hz, 2H), 6.94 (d, *J* = 8.0 Hz, 2H), 6.51 (d, *J* = 8.0 Hz, 2H), 4.95 (s, 2H), 4.16 (d, *J* = 5.8 Hz, 2H), 1.39 (d, *J* = 6.0 Hz, 4H). ^13^C NMR (101 MHz, DMSO-*d*_6_) δ 170.24, 168.24, 158.15 (d, *J* = 240.5 Hz), 147.50, 135.02, 128.16, 126.10, 121.81 (d, *J* = 7.7 Hz), 115.10 (d, *J* = 21.9 Hz), 113.69, 42.35, 29.24, 15.84.

*N-(4-(aminomethyl)phenyl)-N-(3-chlorophenyl)cyclopropane-1,1-dicarboxamide* (**10C**). Brown solid (65%),^1^H NMR (400 MHz, DMSO-*d*_6_) δ 10.83 (s, 1H), 8.28 (t, *J* = 5.8 Hz, 1H), 7.85 (t, *J* = 2.1 Hz, 1H), 7.45 (ddd, *J* = 8.2, 2.1, 1.0 Hz, 1H), 7.32 (t, *J* = 8.1 Hz, 1H), 7.11 (ddd, *J* = 7.9, 2.1, 1.0 Hz, 1H), 6.95–6.88 (m, 2H), 6.52–6.46 (m, 2H), 4.95 (s, 2H), 4.15 (d, *J* = 5.8 Hz, 2H), 1.38 (d, *J* = 2.3 Hz, 4H). ^13^C NMR (101 MHz, DMSO-*d*_6_) δ 170.52, 169.00, 148.00, 140.64, 133.42, 130.71, 128.66, 126.57, 123.60, 119.92, 118.72, 114.15, 42.85, 30.14, 16.35.

*N-(4-(aminomethyl)phenyl)-N-(4-chlorophenyl)cyclopropane-1,1-dicarboxamide*(**10D**). Brown solid (70%),^1^H NMR (400 MHz, DMSO-*d*_6_) δ 10.79 (s, 1H), 8.27 (d, *J* = 5.9 Hz, 1H), 7.63 (d, *J* = 8.4 Hz, 2H), 7.35 (d, *J* = 8.4 Hz, 2H), 6.93 (d, *J* = 8.0 Hz, 2H), 6.50 (d, *J* = 7.9 Hz, 2H), 4.96 (s, 2H), 4.15 (d, *J* = 5.8 Hz, 2H), 1.38 (d, *J* = 4.6 Hz, 4H). ^13^C NMR (101 MHz, DMSO-*d*_6_) δ 170.17, 168.34, 147.47, 137.62, 128.45, 128.15, 127.02, 126.08, 121.46, 113.68, 42.35, 29.43, 15.88.

*N-(4-(aminomethyl)phenyl)-N-(3-bromophenyl)cyclopropane-1,1-dicarboxamide* (**10E**). Brown solid (50%), ^1^H NMR (400 MHz, DMSO-*d*_6_) δ 10.80 (s, 1H), 8.28 (t, *J* = 5.8 Hz, 1H), 7.99 (s, 1H), 7.49 (d, *J* = 6.8 Hz, 1H), 7.25 (d, *J* = 6.5 Hz, 2H), 6.93 (d, *J* = 8.0 Hz, 2H), 6.50 (d, *J* = 7.9 Hz, 2H), 4.96 (s, 2H), 4.14 (d, *J* = 5.7 Hz, 2H), 1.37 (s, 4H). ^13^C NMR (101 MHz, DMSO-*d*_6_) δ 169.99, 168.50, 147.46, 140.27, 130.52, 128.15, 126.11, 126.00, 122.29, 121.39, 118.62, 113.67, 42.36, 29.65, 15.83.

*N-(4-(aminomethyl)phenyl)-N-(4-bromophenyl)cyclopropane-1,1-dicarboxamide* (**10F**). Brown solid (55%),^1^H NMR (400 MHz, DMSO-*d*_6_) δ 10.80 (s, 1H), 8.28 (t, *J* = 5.8 Hz, 1H), 7.58 (d, *J* = 8.6 Hz, 2H), 7.47 (d, *J* = 8.5 Hz, 2H), 6.93 (d, *J* = 7.9 Hz, 2H), 6.50 (d, *J* = 7.9 Hz, 2H), 4.95 (s, 2H), 4.15 (d, *J* = 5.6 Hz, 2H), 1.38 (d, *J* = 4.7 Hz, 4H). ^13^C NMR (101 MHz, DMSO-*d*_6_) δ 170.18, 168.35, 147.49, 138.03, 131.36, 128.16, 126.07, 121.82, 115.05, 113.67, 42.35, 29.44, 15.92.

*N-(4-(aminomethyl)phenyl)-N-(3-methoxyphenyl)cyclopropane-1,1-dicarboxamide*(**10G**). Brown solid (71%),^1^H NMR (400 MHz, DMSO-*d*_6_) δ 10.76 (s, 1H), 8.25 (t, *J* = 5.9 Hz, 1H), 7.29 (s, 1H), 7.19 (t, *J* = 8.1 Hz, 1H), 7.11 (d, *J* = 8.1 Hz, 1H), 6.92 (d, *J* = 8.0 Hz, 2H), 6.64 (dd, *J* = 8.1, 2.4 Hz, 1H), 6.50 (d, *J* = 8.0 Hz, 2H), 4.98 (s, 2H), 4.14 (d, *J* = 5.7 Hz, 2H), 3.72 (s, 3H), 1.38 (d, *J* = 7.8 Hz, 4H). ^13^C NMR (101 MHz, DMSO-*d*_6_) δ 170.48, 168.17, 159.45, 147.42, 139.78, 129.38, 128.13, 126.10, 113.70, 112.05, 109.13, 105.45, 54.97, 42.30, 29.20, 15.98.

*N-(4-(aminomethyl)phenyl)-N-(4-methoxyphenyl)cyclopropane-1,1-dicarboxamide* (**10H**). Brown solid (74%),^1^H NMR (400 MHz, DMSO-*d*_6_) δ 10.54 (s, 1H), 8.36 (t, *J* = 5.4 Hz, 1H), 7.48 (d, *J* = 8.6 Hz, 2H), 6.93 (d, *J* = 8.0 Hz, 2H), 6.87 (d, *J* = 8.6 Hz, 2H), 6.50 (d, *J* = 8.0 Hz, 2H), 4.97 (s, 2H), 4.14 (d, *J* = 5.7 Hz, 2H), 3.71 (s, 3H), 1.38 (d, *J* = 7.7 Hz, 4H). ^13^C NMR (101 MHz, DMSO-*d_6_*) δ 170.48, 167.98,155.42, 147.51, 131.66, 128.15, 126.08, 121.61, 113.68, 55.12, 42.30, 28.88, 15.90.

*N-(4-(aminomethyl)phenyl)-N-(m-tolyl)cyclopropane-1,1-dicarboxamide* (**10I**). Brown solid (62%),^1^H NMR (400 MHz, DMSO-*d*_6_) δ 10.73 (s, 1H), 8.28 (s, 1H), 7.49–7.25 (m, 2H), 7.17 (t, *J* = 7.7 Hz, 1H), 6.51 (d, *J* = 7.9 Hz, 2H), 4.96 (s, 2H), 4.15 (d, *J* = 5.7 Hz, 2H), 2.27 (s, 3H), 1.40 (d, *J* = 8.3 Hz, 4H). ^13^C NMR (101 MHz, DMSO-*d*_6_) δ 170.59, 168.12, 147.51, 138.47, 137.85, 128.46, 128.16, 126.06, 124.19,120.41, 117.01, 113.69, 42.32, 28.97, 21.07, 16.07.

*N-(4-(aminomethyl)phenyl)-N-(p-tolyl)cyclopropane-1,1-dicarboxamide* (**10J**). Brown solid (69%),^1^H NMR (400 MHz, DMSO-*d*_6_) δ 10.67 (s, 1H), 8.29 (d, *J* = 5.9 Hz, 1H), 7.46 (dd, *J* = 8.2, 2.1 Hz, 2H), 7.10 (d, *J* = 7.9 Hz, 2H), 6.94 (d, *J* = 7.8 Hz, 2H), 6.52 (dd, *J* = 8.2, 2.2 Hz, 2H), 4.97 (s, 2H), 4.16 (d, *J* = 5.5 Hz, 2H), 2.25 (s, 3H), 1.40 (dd, *J* = 10.4, 2.7 Hz, 4H). ^13^C NMR (101 MHz, DMSO-*d*_6_) δ 170.56, 168.04, 147.48, 136.05, 132.45, 129.00, 128.16, 126.09, 119.92, 113.71, 42.32, 28.94, 20.41, 16.01.

*N-(4-(aminomethyl)phenyl)-N-(3-cyanophenyl)cyclopropane-1,1-dicarboxamide* (**10K**). Brown solid (59%),^1^H NMR (400 MHz, DMSO-*d*_6_) δ 10.94 (s, 1H), 8.29 (s, 1H), 8.13 (s, 1H), 7.85 – 7.68 (m, 1H), 7.51 (d, *J* = 4.7 Hz, 2H), 6.92 (d, *J* = 7.8 Hz, 2H), 6.50 (d, *J* = 7.9 Hz, 2H), 4.98 (s, 2H), 4.15 (d, *J* = 5.8 Hz, 2H), 1.39 (s, 4H). ^13^C NMR (101 MHz, DMSO-*d*_6_) δ 169.87, 168.77, 147.49, 139.51, 130.01, 128.16, 126.92, 126.08, 124.48, 122.73, 118.67, 113.66, 111.37, 42.37, 29.72, 15.85.

*N-(4-(aminomethyl)phenyl)-N-(4-cyanophenyl)cyclopropane-1,1-dicarboxamide* (**10L**). Brown solid (63%),^1^H NMR (400 MHz, DMSO-*d*_6_) δ 11.12 (s, 1H), 8.29 (s, 1H), 7.81 (d, *J* = 8.9 Hz, 2H), 7.76 (d, *J* = 8.9 Hz, 2H), 6.93 (d, *J* = 8.4 Hz, 2H), 6.60–6.34 (m, 2H), 5.01 (s, 2H), 4.15 (d, *J* = 5.7 Hz, 2H), 1.40 (s, 4H). ^13^C NMR (101 MHz, DMSO-*d*_6_) δ 170.40, 169.29, 147.96, 143.47, 133.56, 128.66, 126.56, 120.28, 119.54, 114.16, 105.54, 42.86, 30.46, 16.44.

*N-(4-(aminomethyl)phenyl)-N-(3-(trifluoromethyl)phenyl)cyclopropane-1,1-dicarboxamide* (**10M**). Brown solid (76%), ^1^H NMR (400 MHz, DMSO-*d*_6_) δ 10.91 (s, 1H), 8.31 (t, *J* = 5.8 Hz, 1H), 8.15 (s, 1H), 7.79 (d, *J* = 8.2 Hz, 1H), 7.53 (t, *J* = 8.0 Hz, 1H), 7.40 (d, *J* = 7.7 Hz, 1H), 6.93 (d, *J* = 7.9 Hz, 2H), 6.50 (d, *J* = 7.9 Hz, 2H), 4.95 (s, 2H), 4.16 (d, *J* = 5.6 Hz, 2H), 1.39 (d, *J* = 4.6 Hz, 4H). ^13^C NMR (101 MHz, DMSO-*d*_6_) δ 169.87, 168.74, 147.49, 139.51, 129.71, 129.30 (d, *J* = 31.7 Hz), 128.15, 126.11, 124.10 (d, *J* = 272.2 Hz), 123.41, 119.68 (d, *J* = 3.8 Hz), 116.05 (d, *J* = 4.4 Hz), 113.65, 42.38, 29.80, 15.75.

*N-(4-(aminomethyl)phenyl)-N-(4-(trifluoromethyl)phenyl)cyclopropane-1,1-dicarboxamide* (**10N**). Brown solid (70%), ^1^H NMR (400 MHz, DMSO-*d*_6_) δ 11.08 (s, 1H), 8.28 (d, *J* = 5.9 Hz, 1H), 7.83 (d, *J* = 8.4 Hz, 2H), 7.66 (d, *J* = 8.4 Hz, 2H), 6.93 (d, *J* = 7.9 Hz, 2H), 6.51 (d, *J* = 7.9 Hz, 2H), 4.95 (s, 2H), 4.16 (d, *J* = 5.6 Hz, 2H), 1.41 (d, *J* = 3.2 Hz, 4H). ^13^C NMR (101 MHz, DMSO-*d*_6_) δ 170.12, 168.71, 147.50, 142.27, 128.16, 126.06, 125.84 (d, *J* = 3.6 Hz), 124.62 (d, *J* = 213.2 Hz), 123.12 (d, *J* = 26.2 Hz), 119.68, 113.66, 42.37, 29.63, 16.02.

*N-(4-(aminomethyl)phenyl)-N-phenylcyclopropane-1,1-dicarboxamide* (**10O**). Brown solid (78%), ^1^H NMR (400 MHz, DMSO-*d*_6_) δ 10.75 (s, 1H), 8.28 (t, *J* = 5.9 Hz, 1H), 7.58 (d, *J* = 8.0 Hz, 2H), 7.30 (t, *J* = 7.8 Hz, 2H), 7.06 (t, *J* = 7.4 Hz, 1H), 6.93 (d, *J* = 7.9 Hz, 2H), 6.51 (d, *J* = 7.9 Hz, 2H), 4.96 (s, 2H), 4.15 (d, *J* = 5.7 Hz, 2H), 1.40 (d, *J* = 8.7 Hz, 4H). ^13^C NMR (101 MHz, DMSO-*d*_6_) δ 170.47, 168.19, 147.50, 138.59, 128.61, 128.15, 126.07, 123.49, 119.89, 113.68, 42.33, 29.14, 15.98.

*N-(4-(aminomethyl)phenyl)-N-(4-chloro-3-(trifluoromethyl)phenyl)cyclopropane-1,1-dicarboxamide* (**10P**). Brown solid (80%),^1^H NMR (400 MHz, DMSO-*d*_6_) δ 10.94 (s, 1H), 8.31 (t, *J* = 5.9 Hz, 1H), 8.25 (d, *J* = 2.3 Hz, 1H), 7.85 (dd, *J* = 8.8, 2.3 Hz, 1H), 7.63 (d, *J* = 8.7 Hz, 1H), 6.93 (d, *J* = 7.9 Hz, 2H), 6.50 (d, *J* = 7.9 Hz, 2H), 4.95 (s, 2H), 4.15 (d, *J* = 5.7 Hz, 2H), 1.38 (s, 4H).^13^C NMR (101 MHz, DMSO-*d*_6_) δ 169.59, 168.83, 147.49, 138.28, 131.79, 128.15, 126.52 (d, *J* = 30.6 Hz), 126.09, 124.62, 123.99, 122.73 (d, *J* = 272.8 Hz), 118.63 (d, *J* = 5.7 Hz), 113.64, 42.39, 30.03, 15.65.

#### 3.1.5. General Procedure for the Preparation of **12A**-**P**

To a suspension of 6-(2-chloro-5-fluoropyrimidin-4-yl)-1-isopropyl-1H-benzo[d](1,2,3)riazole (**6**) (583.4 mg, 2 mmol) in 20 mL 1,4-dioxane, the following compounds were added: **10A–P** (2 mmol), Pd(OAc)_2_ (11 mg, 0.05 mmol), BINAP (62 mg, 0.1 mmol) and Cs_2_CO_3_ (978 mg, 3 mmol) and the flask was purged with Ar. Then, the flask was sealed and the mixture was heated for 12 h at 100℃. The reaction was cooled to RT, the solvent was removed under reduced pressure, and the residue was purified by silica gel column chromatography to obtain **12A–P**.

*N-(4-((5-fluoro-4-(1-isopropyl-1H-benzo[d](1,2,3)triazol-6-yl)pyrimidin-2-yl)amino)benzyl)-N-(3-fluorophenyl)cyclopropane-1,1-dicarboxamide*(**12A**). Light yellow solid; 43% yield; mp 104.9 °C. ^1^H NMR (400 MHz, DMSO-d_6_) δ 10.90 (s, 1H), 9.85 (s, 1H), 8.68 (d, *J* = 3.3 Hz, 1H), 8.51 (s, 1H), 8.44 (t, *J* = 5.9 Hz, 1H), 8.21 (d, *J* = 8.8 Hz, 1H), 8.02 (d, *J* = 8.8 Hz, 1H), 7.74 (d, *J* = 8.5 Hz, 2H), 7.63 (d, *J* = 11.5 Hz, 1H), 7.34—7.26 (m, 2H), 7.22 (d, *J* = 8.4 Hz, 2H), 6.90–6.83 (m, 1H), 5.31 (p, *J* = 6.7 Hz, 1H), 4.29 (d, *J* = 5.7 Hz, 2H), 1.67 (d, *J* = 6.7 Hz, 6H), 1.41 (s, 4H). ^13^C NMR (101 MHz, DMSO) δ 170.88, 168.87, 162.53 (d, *J* = 240.9 Hz), 156.97 (d, *J* = 2.7 Hz), 151.79, 151.03 (d, *J* = 9.6 Hz), 149.28, 148.26 (d, *J* = 25.6 Hz), 146.63, 140.93 (d, *J* = 11.0 Hz), 139.65, 132.87, 132.36, 132.30, 130.63 (d, *J* = 9.5 Hz), 127.95, 124.69 (d, *J* = 5.6 Hz), 119.98, 119.07, 116.02 (d, *J* = 2.5 Hz), 112.23 (d, *J* = 6.6 Hz), 110.34 (d, *J* = 21.1 Hz), 107.15 (d, *J* = 26.2 Hz), 51.76, 42.75, 30.17, 22.47, 16.44. ^19^F NMR (376 MHz, DMSO-d_6_) δ -107.43(s, 1F), -145.82 (s, 1F). ESI-HRMS *m*/*z* calcd. for chemical formula: C_31_H_29_F_2_N_8_O_2_^+^ 583.2376, found 583.2367 [M + H]^+^. HPLC purity 99%.

*N-(4-((5-fluoro-4-(1-isopropyl-1H-benzo[d](1,2,3)triazol-6-yl)pyrimidin-2-yl)amino)benzyl)-N-(4-fluorophenyl)cyclopropane-1,1-dicarboxamide* (**12B**).Light yellow solid; 47% yield; mp 109.5 °C. ^1^H NMR (400 MHz, DMSO-d_6_) δ 10.67 (s, 1H), 9.84 (s, 1H), 8.68 (d, *J* = 3.4 Hz, 1H), 8.51 (s, 1H), 8.45 (t, *J* = 5.9 Hz, 1H), 8.22 (d, *J* = 8.8 Hz, 1H), 8.03 (d, *J* = 8.9 Hz, 1H), 7.76–7.72 (m, 2H), 7.65–7.56 (m, 2H), 7.22 (d, *J* = 8.3 Hz, 2H), 7.12 (t, *J* = 8.9 Hz, 2H), 5.32 (p, *J* = 6.7 Hz, 1H), 4.29 (d, *J* = 5.8 Hz, 2H), 1.68 (d, *J* = 6.7 Hz, 6H), 1.41 (d, *J* = 2.4 Hz, 4H). ^13^C NMR (101 MHz, DMSO) δ 170.45, 168.14, 158.14 (d, *J* = 239.8 Hz), 156.89–156.32 (m), 151.30, 150.54 (d, *J* = 8.7 Hz), 148.79, 147.76 (d, *J* = 26.1 Hz), 146.14, 139.15, 135.02 (d, *J* = 2.1 Hz), 132.40, 131.87, 127.45, 124.19 (d, *J* = 5.5 Hz), 121.80 (d, *J* = 7.7 Hz), 119.49, 118.61, 115.20, 114.97, 111.73 (d, *J* = 6.5 Hz), 51.27, 42.25, 29.38, 21.97, 15.85. ^19^F NMR (376 MHz, DMSO-d_6_) δ −114.18(s, 1F), −145.83 (s, 1F). ESI-HRMS *m*/*z* calcd. for chemical formula: C_31_H_29_F_2_N_8_O_2_^+^ 583.2376, found 583.2377 [M + H]^+^. HPLC purity 98%.

*N-(3-chlorophenyl)-N-(4-((5-fluoro-4-(1-isopropyl-1H-benzo[d](1,2,3)triazol-6-yl)pyrimidin-2-yl)amino)benzyl)cyclopropane-1,1-dicarboxamide* (**12C**). Light yellow solid; 56% yield; mp 119.4 °C.^1^H NMR (400 MHz, DMSO-d_6_) δ 10.83 (s, 1H), 9.85 (s, 1H), 8.68 (d, *J* = 3.4 Hz, 1H), 8.51 (s, 1H), 8.45 (t, *J* = 5.9 Hz, 1H), 8.22 (d, *J* = 8.8 Hz, 1H), 8.02 (d, *J* = 8.8 Hz, 1H), 7.85 (s, 1H), 7.74 (d, *J* = 8.5 Hz, 2H), 7.46 (d, *J* = 8.3 Hz, 1H), 7.30 (t, *J* = 8.1 Hz, 1H), 7.22 (d, *J* = 8.3 Hz, 2H), 7.09 (dd, *J* = 7.8, 2.2 Hz, 1H), 5.32 (p, *J* = 6.7 Hz, 1H), 4.29 (d, *J* = 5.7 Hz, 2H), 1.68 (d, *J* = 6.7 Hz, 6H), 1.40 (s, 4H). ^13^C NMR (101 MHz, DMSO) δ 170.75, 168.92, 157.07–156.90 (m), 151.79, 151.04 (d, *J* = 9.7 Hz), 149.28, 148.26 (d, *J* = 26.1 Hz), 146.63, 140.65, 139.65, 133.41, 132.88, 132.36, 132.30, 130.69, 127.96, 124.70 (d, *J* = 5.9 Hz), 123.60, 119.99, 119.92, 119.07, 118.71, 112.23 (d, *J* = 6.6 Hz), 51.77, 42.77, 30.26, 22.48, 16.38. ^19^F NMR (376 MHz, DMSO-d_6_) δ −145.8 (s, 1F). ESI-HRMS *m*/*z* calcd. for chemical formula: C_31_H_29_ClFN_8_O_2_^+^ 599.2081, found 599.2077 [M + H]^+^. HPLC purity 99%.

*N-(4-chlorophenyl)-N-(4-((5-fluoro-4-(1-isopropyl-1H-benzo[d](1,2,3)triazol-6-yl)pyrimidin-2-yl)amino)benzyl)cyclopropane-1,1-dicarboxamide* (**12D**). Light yellow solid; 52% yield; mp118 °C. ^1^H NMR (400 MHz, DMSO-d_6_) δ 10.79 (s, 1H), 9.84 (s, 1H), 8.67 (d, *J* = 3.4 Hz, 1H), 8.51–8.49 (m, 1H), 8.44 (t, *J* = 5.9 Hz, 1H), 8.21 (d, *J* = 8.8 Hz, 1H), 8.02 (d, *J* = 8.8 Hz, 1H), 7.73 (d, *J* = 8.6 Hz, 2H), 7.63 (d, *J* = 8.9 Hz, 2H), 7.32 (d, *J* = 8.9 Hz, 2H), 7.21 (d, *J* = 8.6 Hz, 2H), 5.31 (p, *J* = 6.7 Hz, 1H), 4.28 (d, *J* = 5.8 Hz, 2H), 1.67 (d, *J* = 6.7 Hz, 6H), 1.47–1.32 (m, 4H). ^13^C NMR (101 MHz, DMSO) δ 170.91, 168.73, 156.98 (d, *J* = 2.4 Hz), 151.79, 151.04 (d, *J* = 9.2 Hz), 149.28, 148.25 (d, *J* = 25.5 Hz), 146.63, 139.65, 138.13, 132.87, 132.36, 132.30, 128.93, 127.96, 127.51, 124.70 (d, *J* = 6.0 Hz), 121.94, 119.98, 119.09, 112.23 (d, *J* = 6.7 Hz), 51.76, 42.75, 30.06, 22.47, 16.42. ^19^F NMR (376 MHz, DMSO-d_6_) δ −145.81 (s, 1F). ESI-HRMS *m*/*z* calcd. for chemical formula: C_31_H_29_ClFN_8_O_2_^+^ 599.2081, found 599.2072 [M + H]^+^. HPLC purity 96%.

*N-(3-bromophenyl)-N-(4-((5-fluoro-4-(1-isopropyl-1H-benzo[d](1,2,3)triazol-6-yl)pyrimidin-2-yl)amino)benzyl)cyclopropane-1,1-dicarboxamide* (**12E**). Light yellow solid; 44% yield; mp 114.6 °C. ^1^H NMR (400 MHz, DMSO-d_6_) δ 10.80 (s, 1H), 9.85 (s, 1H), 8.68 (d, *J* = 3.4 Hz, 1H), 8.51 (s, 1H), 8.45 (t, *J* = 5.9 Hz, 1H), 8.22 (d, *J* = 8.8 Hz, 1H), 8.08-7.96 (m, 2H), 7.74 (d, *J* = 8.4 Hz, 2H), 7.49 (d, *J* = 2.2 Hz, 1H), 7.27–7.16 (m, 4H), 5.32 (p, *J* = 6.7 Hz, 1H), 4.29 (d, *J* = 5.8 Hz, 2H), 1.68 (d, *J* = 6.7 Hz, 6H), 1.40 (s, 4H). ^13^C NMR (101 MHz, DMSO) δ 170.72, 168.91, 156.98 (d, *J* = 3.0 Hz), 151.79, 151.03 (d, *J* = 9.3 Hz), 149.28, 148.26 (d, *J* = 25.9 Hz), 146.63, 140.79, 139.65, 132.88, 132.36, 132.31, 131.00, 127.96, 126.50, 124.70 (d, *J* = 5.9 Hz), 122.78, 121.88, 119.99, 119.10, 119.08, 112.23 (d, *J* = 6.7 Hz), 51.77, 42.77, 30.29, 22.48, 16.36. ^19^F NMR (376 MHz, DMSO-d_6_) δ −145.81 (s, 1F). ESI-HRMS *m*/*z* calcd. for chemical formula: C_31_H_29_BrFN_8_O_2_^+^ 643.1575, found 643.1572 [M + H]^+^. HPLC purity 97%.

N-(4-bromophenyl)-N-(4-((5-fluoro-4-(1-isopropyl-1H-benzo[d](1,2,3)triazol-6-yl)pyrimidin-2-yl)amino)benzyl)cyclopropane-1,1-dicarboxamide (**12F**). Light yellow solid; 42% yield; mp 214 °C. ^1^H NMR (400 MHz, DMSO-d_6_) δ 10.79 (s, 1H), 9.84 (s, 1H), 8.68 (d, *J* = 3.5 Hz, 1H), 8.51 (s, 1H), 8.44 (t, *J* = 5.9 Hz, 1H), 8.22 (d, *J* = 8.7 Hz, 1H), 8.02 (d, *J* = 8.8 Hz, 1H), 7.74 (d, *J* = 8.2 Hz, 2H), 7.58 (d, *J* = 9.0 Hz, 2H), 7.46 (d, *J* = 8.9 Hz, 2H), 7.21 (d, *J* = 8.3 Hz, 2H), 5.32 (p, *J* = 6.8 Hz, 1H), 4.29 (d, *J* = 5.7 Hz, 2H), 1.68 (d, *J* = 6.8 Hz, 6H), 1.41 (s, 4H). ^13^C NMR (101 MHz, DMSO) δ 170.40, 168.25, 156.48 (d, *J* = 2.4 Hz), 151.30, 150.56 (d, *J* = 9.4 Hz), 148.79, 147.77 (d, *J* = 26.1 Hz), 146.13, 139.15, 138.06, 132.37, 131.87, 131.81, 131.35, 127.46, 124.20 (d, *J* = 5.6 Hz), 121.81, 119.50, 118.60, 115.04, 111.74 (d, *J* = 6.8 Hz), 51.27, 42.26, 29.60, 21.99, 15.93. ^19^F NMR (376 MHz, DMSO-d_6_) δ −145.81 (s, 1F). ESI-HRMS *m*/*z* calcd. for chemical formula: C_31_H_29_BrFN_8_O_2_^+^ 643.1575, found 643.1576 [M + H]^+^. HPLC purity 95%.

*N-(4-((5-fluoro-4-(1-isopropyl-1H-benzo[d](1,2,3)triazol-6-yl)pyrimidin-2-yl)amino)benzyl)-N-(3-methoxyphenyl)cyclopropane-1,1-dicarboxamide* (**12G**).Light yellow solid; 47% yield; mp 101 °C. ^1^H NMR (400 MHz, DMSO-d_6_) δ 10.76 (s, 1H), 9.84 (s, 1H), 8.68 (d, *J* = 3.3 Hz, 1H), 8.51 (s, 1H), 8.42 (t, *J* = 5.9 Hz, 1H), 8.22 (d, *J* = 8.9 Hz, 1H), 8.03 (dd, *J* = 8.8, 1.4 Hz, 1H), 7.81–7.64 (m, 2H), 7.31 (d, *J* = 2.3 Hz, 1H), 7.26–7.09 (m, 4H), 6.63 (ddd, *J* = 8.1, 2.7, 1.2 Hz, 1H), 5.32 (p, *J* = 6.7 Hz, 1H), 4.29 (d, *J* = 5.7 Hz, 2H), 3.71 (s, 3H), 1.68 (d, *J* = 6.7 Hz, 6H), 1.42 (d, *J* = 3.9 Hz, 4H). ^13^C NMR (101 MHz, DMSO) δ 170.71, 168.08, 159.44, 156.48 (d, *J* = 2.3 Hz), 151.30, 150.54 (d, *J* = 8.9 Hz), 148.79, 147.76 (d, *J* = 25.8 Hz), 146.14, 139.80, 139.16, 132.37, 131.84 (d, *J* = 5.4 Hz), 129.36, 128.12, 127.44, 124.19 (d, *J* = 5.6 Hz), 119.49, 118.60, 113.67, 112.06, 111.73 (d, *J* = 6.6 Hz), 109.10, 105.49, 54.95, 51.28, 42.23, 29.33, 21.97, 16.03. ^19^F NMR (376 MHz, DMSO-d_6_) δ -145.79 (s, 1F). ESI-HRMS *m*/*z* calcd. for chemical formula: C_32_H_32_FN_8_O_3_^+^ 595.2576, found 595.2578 [M + H]^+^. HPLC purity 99%.

*N-(4-((5-fluoro-4-(1-isopropyl-1H-benzo[d](1,2,3)triazol-6-yl)pyrimidin-2-yl)amino)benzyl)-N-(4-methoxyphenyl)cyclopropane-1,1-dicarboxamide* (**12H**). Light yellow solid; 55% yield; mp 181 °C. ^1^H NMR (400 MHz, DMSO-d_6_) δ 10.50 (s, 1H), 9.84 (s, 1H), 8.68 (d, *J* = 3.3 Hz, 1H), 8.51 (s, 1H), 8.47 (t, *J* = 5.9 Hz, 1H), 8.22 (d, *J* = 8.7 Hz, 1H), 8.03 (d, *J* = 8.8 Hz, 1H), 7.74 (d, *J* = 8.6 Hz, 2H), 7.49 (d, *J* = 9.0 Hz, 2H), 7.22 (d, *J* = 8.6 Hz, 2H), 6.86 (d, *J* = 9.0 Hz, 2H), 5.31 (p, *J* = 6.7 Hz, 1H), 4.29 (d, *J* = 6.0 Hz, 2H), 3.70 (s, 3H), 1.68 (d, *J* = 6.5 Hz, 6H), 1.40 (d, *J* = 6.4 Hz, 4H). ^13^C NMR (101 MHz, DMSO) δ 170.68, 167.84, 156.48 (d, *J* = 2.4 Hz), 155.42, 151.30, 150.53 (d, *J* = 9.1 Hz), 148.79, 147.78 (d, *J* = 25.9 Hz), 146.14, 139.16, 132.39, 131.87, 131.81, 131.69, 127.45, 124.20 (d, *J* = 6.0 Hz), 121.59, 119.49, 118.61, 113.67, 111.73 (d, *J* = 6.8 Hz), 55.10, 51.28, 42.23, 29.04, 21.97, 15.90. ^19^F NMR (376 MHz, DMSO-d_6_) δ −145.79 (s, 1F). ESI-HRMS *m*/*z* calcd. for chemical formula: C_32_H_32_FN_8_O_3_^+^ 595.2576, found 595.2576 [M + H]^+^. HPLC purity 97%.

*N-(4-((5-fluoro-4-(1-isopropyl-1H-benzo[d](1,2,3)triazol-6-yl)pyrimidin-2-yl)amino)benzyl)-N-(m-tolyl)cyclopropane-1,1-dicarboxamide* (**12I**). Light yellow solid; 61% yield; mp 85 °C. ^1^H NMR (400 MHz, DMSO-d_6_) δ 10.72 (s, 1H), 9.85 (s, 1H), 8.68 (d, *J* = 3.4 Hz, 1H), 8.52 (s, 1H), 8.44 (t, *J* = 5.9 Hz, 1H), 8.22 (d, *J* = 8.8 Hz, 1H), 8.03 (dt, *J* = 8.8, 1.4 Hz, 1H), 7.83–7.72 (m, 2H), 7.46–7.36 (m, 2H), 7.22 (d, *J* = 8.5 Hz, 2H), 7.16 (t, *J* = 7.8 Hz, 1H), 6.86 (d, *J* = 7.5 Hz, 1H), 5.32 (p, *J* = 6.7 Hz, 1H), 4.29 (d, *J* = 5.8 Hz, 2H), 2.25 (s, 3H), 1.68 (d, *J* = 6.7 Hz, 6H), 1.42 (d, *J* = 3.8 Hz, 4H). ^13^C NMR (101 MHz, DMSO) δ 171.31, 168.52, 156.98 (d, *J* = 2.1 Hz), 151.80, 151.03 (d, *J* = 9.4 Hz), 149.29, 148.27 (d, *J* = 25.8 Hz), 146.64, 139.67, 138.99, 138.32, 132.86, 132.37, 132.32, 128.93, 127.95, 124.70 (d, *J* = 5.3 Hz), 120.90, 119.99, 119.10, 117.50, 112.23 (d, *J* = 6.6 Hz), 51.77, 42.73, 29.60, 22.46, 21.55, 16.60. ^19^F NMR (376 MHz, DMSO-d_6_) δ −145.81 (s, 1F). ESI-HRMS *m*/*z* calcd. for chemical formula: C_32_H_32_FN_8_O_2_^+^ 579.2627, found 579.2626 [M + H]^+^. HPLC purity98%.

*N-(4-((5-fluoro-4-(1-isopropyl-1H-benzo[d](1,2,3)triazol-6-yl)pyrimidin-2-yl)amino)benzyl)-N-(p-tolyl)cyclopropane-1,1-dicarboxamide* (**12J**). Light yellow solid; 57% yield; mp 190 °C. ^1^H NMR (400 MHz, DMSO-d_6_) δ 10.65 (s, 1H), 9.84 (s, 1H), 8.67 (d, *J* = 3.3 Hz, 1H), 8.51 (s, 1H), 8.45 (t, *J* = 5.9 Hz, 1H), 8.21 (d, *J* = 8.8 Hz, 1H), 8.02 (d, *J* = 8.8 Hz, 1H), 7.75 (d, *J* = 8.3 Hz, 2H), 7.46 (d, *J* = 8.1 Hz, 2H), 7.22 (d, *J* = 8.3 Hz, 2H), 7.07 (d, *J* = 8.1 Hz, 2H), 5.31 (p, *J* = 6.7 Hz, 1H), 4.29 (d, *J* = 5.8 Hz, 2H), 2.22 (s, 3H), 1.67 (d, *J* = 6.7 Hz, 6H), 1.48–1.33 (m, 4H). ^13^C NMR (101 MHz, DMSO) δ 171.29, 168.44, 156.98 (d, *J* = 2.8 Hz), 151.80, 150.99 (d, *J* = 8.7 Hz), 149.29, 148.26 (d, *J* = 25.8 Hz), 146.64, 139.67, 136.56, 132.93, 132.86, 132.36, 132.32, 129.46, 127.94, 124.69 (d, *J* = 5.8 Hz), 120.41, 119.98, 119.11, 112.21 (d, *J* = 6.6 Hz), 51.77, 42.73, 29.56, 22.46, 20.88, 16.54. ^19^F NMR (376 MHz, DMSO-d_6_) δ −145.76 (s, 1F). ESI-HRMS *m*/*z* calcd. for chemical formula: C_32_H_32_FN_8_O_2_^+^ 579.2627, found 579.2623 [M + H]^+^. HPLC purity 95%.

*N-(3-cyanophenyl)-N-(4-((5-fluoro-4-(1-isopropyl-1H-benzo[d](1,2,3)triazol-6-yl)pyrimidin-2-yl)amino)benzyl)cyclopropane-1,1-dicarboxamide* (**12K**). Light yellow solid; 63% yield; mp 108.4 °C. ^1^H NMR (400 MHz, DMSO-d_6_) δ 10.94 (s, 1H), 9.84 (s, 1H), 8.67 (d, *J* = 3.3 Hz, 1H), 8.51 (s, 1H), 8.46 (t, *J* = 5.9 Hz, 1H), 8.21 (d, *J* = 8.8 Hz, 1H), 8.14 (s, 1H), 8.02 (d, *J* = 8.8 Hz, 1H), 7.83 (ddd, *J* = 5.2, 4.1, 2.2 Hz, 1H), 7.74 (d, *J* = 8.6 Hz, 2H), 7.53–7.47 (m, 2H), 7.21 (d, *J* = 8.6 Hz, 2H), 5.32 (p, *J* = 6.7 Hz, 1H), 4.29 (d, *J* = 5.8 Hz, 2H), 1.68 (d, *J* = 6.7 Hz, 6H), 1.41 (s, 4H). ^13^C NMR (101 MHz, DMSO) δ 170.59, 169.18, 156.97 (d, *J* = 2.2 Hz), 151.79, 151.05 (d, *J* = 9.4 Hz), 149.28, 148.25 (d, *J* = 26.1 Hz), 146.63, 140.02, 139.65, 132.87, 132.36, 132.30, 130.50, 127.96, 127.42, 124.96, 124.70 (d, *J* = 5.9 Hz), 123.21, 119.99, 119.15, 119.08, 112.23 (d, *J* = 7.0 Hz), 111.86, 51.76, 42.78, 30.37, 22.48, 16.36. ^19^F NMR (376 MHz, DMSO-d_6_) δ −145.81 (s, 1F). ESI-HRMS *m*/*z* calcd. for chemical formula: C_32_H_29_FN_9_O_2_^+^ 590.2423, found 590.2418 [M + H]^+^. HPLC purity 99%.

*N-(4-cyanophenyl)-N-(4-((5-fluoro-4-(1-isopropyl-1H-benzo[d](1,2,3)triazol-6-yl)pyrimidin-2-yl)amino)benzyl)cyclopropane-1,1-dicarboxamide* (**12L**). Light yellow solid; 59% yield; mp 241 °C. ^1^H NMR (400 MHz, DMSO-d_6_) δ 11.10 (s, 1H), 9.83 (s, 1H), 8.67 (d, *J* = 3.4 Hz, 1H), 8.51 (s, 1H), 8.43 (t, *J* = 5.9 Hz, 1H), 8.21 (d, *J* = 8.8 Hz, 1H), 8.02 (d, *J* = 8.9 Hz, 1H), 7.81 (d, *J* = 8.9 Hz, 2H), 7.77–7.71 (m, 4H), 7.21 (d, *J* = 8.6 Hz, 2H), 5.32 (p, *J* = 6.7 Hz, 1H), 4.29 (d, *J* = 5.7 Hz, 2H), 1.68 (d, *J* = 6.7 Hz, 6H), 1.42 (s, 4H). ^13^C NMR (101 MHz, DMSO) δ 170.64, 169.21, 156.98 (d, *J* = 2.3 Hz), 151.80, 151.05 (d, *J* = 9.4 Hz), 149.29, 148.24 (d, *J* = 25.6 Hz), 146.63, 143.47, 139.65, 133.54, 132.85, 132.36, 132.30, 127.96, 124.69 (d, *J* = 6.0 Hz), 120.30, 119.99, 119.50, 119.09, 112.23 (d, *J* = 6.6 Hz), 105.57, 51.77, 42.78, 30.57, 22.48, 16.46. ^19^F NMR (376 MHz, DMSO-d_6_) δ −145.82 (s, 1F). ESI-HRMS *m*/*z* calcd. for chemical formula: C_32_H_29_FN_9_O_2_^+^ 590.2423, found 590.2420 [M + H]^+^. HPLC purity 96%.

*N-(4-((5-fluoro-4-(1-isopropyl-1H-benzo[d](1,2,3)triazol-6-yl)pyrimidin-2-yl)amino)benzyl)-N-(3-(trifluoromethyl)phenyl)cyclopropane-1,1-dicarboxamide* (**12M**). Light yellow solid; 46% yield; mp 189 °C. ^1^H NMR (400 MHz, DMSO-d_6_) δ 10.90 (s, 1H), 9.84 (s, 1H), 8.68 (d, *J* = 3.4 Hz, 1H), 8.51 (d, *J* = 1.3 Hz, 1H), 8.47 (t, *J* = 5.9 Hz, 1H), 8.22 (d, *J* = 8.8 Hz, 1H), 8.15 (s, 1H), 8.02 (d, *J* = 8.9 Hz, 1H), 7.79 (d, *J* = 8.4 Hz, 1H), 7.74 (d, *J* = 8.6 Hz, 2H), 7.52 (t, *J* = 8.0 Hz, 1H), 7.39 (d, *J* = 7.6 Hz, 1H), 7.22 (d, *J* = 8.7 Hz, 2H), 5.32 (p, *J* = 6.7 Hz, 1H), 4.30 (d, *J* = 5.8 Hz, 2H), 1.68 (d, *J* = 6.7 Hz, 6H), 1.41 (d, *J* = 1.9 Hz, 4H). ^13^C NMR (101 MHz, DMSO) δ 170.58, 169.16, 156.99 (d, *J* = 2.7 Hz), 151.79, 151.06 (d, *J* = 9.1 Hz), 149.29, 148.25 (d, *J* = 25.3 Hz), 146.64, 140.02, 139.65, 132.91, 132.36, 132.32, 130.21, 129.78 (d, *J* = 31.4 Hz), 127.96, 125.95, 124.69 (d, *J* = 6.0 Hz), 123.91, 120.20 (d, *J* = 4.2 Hz), 119.99, 119.09, 116.58 (d, *J* = 7.1 Hz), 112.23 (d, *J* = 6.6 Hz), 51.76, 42.79, 30.44, 22.46, 16.27. ^19^F NMR (376 MHz, DMSO-d_6_) δ -56.5(s, 3F), −145.86 (s, 1F). ESI-HRMS *m*/*z* calcd. for chemical formula: C_32_H_29_F_4_N_8_O_2_^+^ 633.2344, found 633.2347 [M + H]^+^. HPLC purity 99%.

*N-(4-((5-fluoro-4-(1-isopropyl-1H-benzo[d](1,2,3)triazol-6-yl)pyrimidin-2-yl)amino)benzyl)-N-(4-(trifluoromethyl)phenyl)cyclopropane-1,1-dicarboxamide* (**12N**). Light yellow solid; 48% yield; mp 219 °C. ^1^H NMR (400 MHz, DMSO-d_6_) δ 11.06 (s, 1H), 9.84 (s, 1H), 8.68 (d, *J* = 3.3 Hz, 1H), 8.51 (d, *J* = 1.3 Hz, 1H), 8.44 (t, *J* = 5.8 Hz, 1H), 8.22 (dd, *J* = 8.7, 0.8 Hz, 1H), 8.02 (dt, *J* = 8.8, 1.3 Hz, 1H), 7.83 (d, *J* = 8.4 Hz, 2H), 7.78 – 7.71 (m, 2H), 7.64 (d, *J* = 8.6 Hz, 2H), 7.26–7.17 (m, 2H), 5.32 (p, *J* = 6.7 Hz, 1H), 4.29 (d, *J* = 5.8 Hz, 2H), 1.68 (d, *J* = 6.7 Hz, 6H), 1.43 (s, 4H). ^13^C NMR (101 MHz, DMSO) δ 170.82, 169.11, 156.99 (d, *J* = 2.6 Hz), 151.79, 151.06 (d, *J* = 9.7 Hz), 149.29, 148.25 (d, *J* = 25.6 Hz), 146.64, 142.78, 139.66, 132.86, 132.36, 132.30, 127.96, 126.34 (d, *J* = 4.5 Hz), 126.17, 124.69 (d, *J* = 6.0 Hz), 124.12–123.42 (m), 120.19, 119.99, 119.09, 112.23 (d, *J* = 6.6 Hz), 51.76, 42.77, 30.30, 22.47, 16.51. ^19^F NMR (376 MHz, DMSO-d_6_) δ −55.52(s, 3F), −145.85 (s, 1F). ESI-HRMS *m*/*z* calcd. for chemical formula: C_32_H_29_F_4_N_8_O_2_^+^ 633.2344, found 633.2343 [M + H]^+^. HPLC purity 98%.

*N-(4-((5-fluoro-4-(1-isopropyl-1H-benzo[d](1,2,3)triazol-6-yl)pyrimidin-2-yl)amino)benzyl)-N-phenylcyclopropane-1,1-dicarboxamide* (**12O**). Light yellow solid; 41% yield; mp 103 °C. ^1^H NMR (400 MHz, DMSO-d_6_) δ 10.74 (s, 1H), 9.85 (s, 1H), 8.68 (dd, *J* = 3.4, 1.0 Hz, 1H), 8.51 (d, *J* = 1.3 Hz, 1H), 8.44 (t, *J* = 5.9 Hz, 1H), 8.22 (d, *J* = 8.8 Hz, 1H), 8.02 (dt, *J* = 8.8, 1.3 Hz, 1H), 7.84–7.71 (m, 2H), 7.65–7.54 (m, 2H), 7.28 (dd, *J* = 8.5, 7.3 Hz, 2H), 7.24–7.18 (m, 2H), 7.09–6.97 (m, 1H), 5.31 (p, *J* = 6.7 Hz, 1H), 4.29 (d, *J* = 5.8 Hz, 2H), 1.67 (d, *J* = 6.7 Hz, 6H), 1.45–1.35 (m, 4H). ^13^C NMR (101 MHz, DMSO) δ 171.20, 168.59, 156.98 (d, *J* = 2.0 Hz), 151.79, 151.03 (d, *J* = 9.8 Hz), 149.28, 148.28 (d, *J* = 26.2 Hz), 146.63, 139.65, 139.11, 132.87, 132.36, 132.31, 129.08, 127.95, 124.70 (d, *J* = 5.8 Hz), 123.98, 120.38, 119.99, 119.09, 112.24 (d, *J* = 6.3 Hz), 51.77, 42.73, 29.77, 22.47, 16.52. ^19^F NMR (376 MHz, DMSO-d_6_) δ −145.81 (s, 1F). ESI-HRMS *m*/*z* calcd. for chemical formula: C_31_H_30_FN_8_O_2_^+^ 565.2470, found 565.2468 [M + H]^+^. HPLC purity 99%.

*N-(4-chloro-3-(trifluoromethyl)phenyl)-N-(4-((5-fluoro-4-(1-isopropyl-1H-benzo[d](1,2,3)triazol-6-yl)pyrimidin-2-yl)amino)benzyl)cyclopropane-1,1-dicarboxamide* (**12P**). Light yellow solid; 44% yield; mp 108.1 °C. ^1^H NMR (400 MHz, DMSO-d_6_) δ 10.93 (s, 1H), 9.84 (s, 1H), 8.67 (d, *J* = 3.3 Hz, 1H), 8.51 (d, *J* = 1.3 Hz, 1H), 8.48 (t, *J* = 5.9 Hz, 1H), 8.25 (d, *J* = 2.6 Hz, 1H), 8.22 (dd, *J* = 8.8, 0.7 Hz, 1H), 8.02 (dt, *J* = 8.8, 1.4 Hz, 1H), 7.85 (dd, *J* = 8.8, 2.6 Hz, 1H), 7.73 (d, *J* = 8.6 Hz, 2H), 7.62 (d, *J* = 8.8 Hz, 1H), 7.21 (d, *J* = 8.6 Hz, 2H), 5.32 (p, *J* = 6.7 Hz, 1H), 4.29 (d, *J* = 5.8 Hz, 2H), 1.68 (d, *J* = 6.7 Hz, 6H), 1.40 (s, 4H). ^13^C NMR (101 MHz, DMSO) δ 170.29, 169.23, 156.97 (d, *J* = 2.2 Hz), 151.79, 151.06 (d, *J* = 8.3 Hz), 149.28, 148.24 (d, *J* = 26.2 Hz), 146.63, 139.65, 138.80, 132.89, 132.36, 132.30, 127.97, 126.99 (d, *J* = 31.0 Hz), 125.11, 124.68 (d, *J* = 5.8 Hz), 124.53 (d, *J* = 8.3 Hz), 121.85, 119.98, 119.14, 119.07, 112.22 (d, *J* = 6.3 Hz), 51.76, 42.80, 30.69, 22.47, 16.15. ^19^F NMR (376 MHz, DMSO-d_6_) δ −56.74(s, 3F), −145.88 (s, 1F). ESI-HRMS m/z calcd. for chemical formula: C_32_H_28_ClF_4_N_8_O_2_^+^ 667.1954, found 667.1953 [M + H]^+^. HPLC purity 96%.

### 3.2. Biology

#### 3.2.1. Molecular Docking

All the calculations were carried out using the platform of Discovery Studio 3.1 (DS 3.1, Accelrys Inc., San Diego, CA). The receptor protein was prepared by the DS 3.1 software package. Docking studies were finished by Discovery Studio 3.1 to explore the predicted binding modes of compound **12O** in CDK2 (PDB entry: 1H1P), CDK5 (PDB entry: 1UNG), CDK9 (PDB entry: 4BCF), respectively. Hydrogen atoms were added to proteins using Discovery Studio 3.1. The images were created by PyMOL.

#### 3.2.2. Kinase Inhibition Assays

Kinase inhibition profiles were determined using KinaseProfiler services provided by Eurofins, and ATP concentrations used are the Km of corresponding kinases.

CDK1/cyclinB (h) (CDK2/cyclinA (h), CDK3/cyclinE (h), CDK4/CyclinD(h), CDK5/p25 (h), CDK6/cyclinD3 (h)) were incubated with 8 mM MOPS pH 7.0, 0.2 mM EDTA, 0.1 mg/mL histone H1, 10 mM MgAcetate and [gamma-33P]-ATP (specific activity and concentration as required).

CDK7/cyclinH/MAT1 (h) was incubated with 8 mM MOPS pH 7.0, 0.2 mM EDTA, 500 uM peptide, 10 mM MgAcetate and [gamma-33P]-ATP (specific activity and concentration as required).

CDK9/cyclinT1 (h) was incubated with 8 mM MOPS pH 7.0, 0.2 mM EDTA, 100 uM KTFCGTPEYLAPEVRREPRILSEEEQEMFRDFDYIADWC, 10 mM MgAcetate and [gamma-33P]-ATP (specific activity and concentration as required).

Flt1 (h) was incubated with 8 mM MOPS pH 7.0, 0.2 mM EDTA, 250 uM KKKSPGEYVNIEFG, 10 mM MgAcetate and [g- 33P]-ATP; Flt3 (h) [Abl (h)] was incubated with 8 mM MOPS pH 7.0, 0.2 mM EDTA, 50 uM EAIYAAPFAKKK, 10 mM MgAcetate and [gamma-33P]- ATP; Flt4 (h) was incubated with 8 mM MOPS pH 7.0, 0.2 mM EDTA, 500 uM GGEEEEYFELVKKKK, 10 mM MgAcetate and [gamma-33P]-ATP.

c-Kit (h) was incubated with 8 mM MOPS pH 7.0, 0.2 mM EDTA, 10 mM MnCl2, 0.1 mg/mL poly(Glu, Tyr) 4:1, 10 mM MgAcetate and [gamma-33P]-ATP.

FGFR1 (h) was incubated with 8 mM MOPS pH 7.0, 0.2 mM EDTA, 250 uM KKKSPGEYVNIEFG, 10 mM MgAcetate and [gamma-33P]-ATP.

Src (1-530) (h) was incubated with 8 mM MOPS pH 7.0, 0.2 mM EDTA, 500 uM GGEEEEYFELVKKKK, 10 mM MgAcetate and [gamma-33P]-ATP.

KDR (h) was incubated with 8 mM MOPS pH 7.0, 0.2 mM EDTA, 0.33 mg/mL myelin basic protein, 10 mM MgAcetate and [gamma-33P]-ATP.

PDGFR-alfa(h) [PDGFR-beta(h)] was incubated with 8 mM MOPS pH 7.0, 0.2 mM EDTA, 0.1 mg/mL poly(Glu, Tyr) 4:1, 10 mM MnCl2, 10 mM MgAcetate and [gamma-33P]-ATP.

All the reactions above were initiated by the addition of the Mg/ATP mix. After incubation for 40 min at room temperature, the reaction was stopped by the addition of phosphoric acid to a concentration of 0.5%. An amount of 10 µL of the reaction was then spotted onto a P30 filtermat and washed four times for 4 min in 0.425% phosphoric acid and once in methanol prior to drying and scintillation counting.

#### 3.2.3. Cytotoxicity Assay

The viability of cells as determined using the CCK8 assay method. Cells were seeded (2000–3000 cells per well) in 96-well plates. After incubation for 24 h in serum-containing media, the cells were treated with drugs at different concentrations diluted with culture medium for 72 h at 37 °C with a 5% CO_2_ atmosphere. Then, 10 μL CCK8 regent (Promega, WI) was added to each well, and the plates were incubated for 1–4 h at 37 °C. Finally, absorbance values of test wells (A_S_), control wells (A_C_) and blank wells (A_b_) at 450 nm were read using the Microplate Reader (Promega, WI). Inhibition ratios were calculated as follows: [(A_C_ − A_S_)/ (A_C_−A_b_)] × 100%. IC_50_ values were calculated using GraphPad software [23].

#### 3.2.4. Cell Cycle Assay

Cells were plated on 6-well culture plates at a density of 5 × 10^5^ cells/mL. The cells were treated with the indicated concentrations of **12O** or ribociclib for 24 h after their adherence. Cells were washed with PBS for three times and then fixed with ice cold 75% ethanol overnight. The fixed cells were then washed with PBS and stained with propidium iodide (50 mg/mL) in the presence of RNase A (0.5 mg) for 30 min at 37 °C. The stained cells were then subjected to flow cytometry (Becton Dickinson (BD), San Jose, CA, USA.) for cell cycle analysis.

#### 3.2.5. Annexin V-FITC/ PI Apoptosis Assay

Cells at a density of 3 × 10^5^ cells/mL were seeded in 6-well plates and treated with compounds at different concentrations for 48 h. The cells were then harvested and washed twice with cold PBS. Then, the cells were subjected to an Annexin V/PI Apoptosis Detection kit (BD Biosciences) for staining according to manufacturer’s instructions, and finally analyzed by flow cytometry (Modfit, BD).

#### 3.2.6. Colony Formation Assay

Cells treated with indicated compounds or vehicle were washed with PBS, trypsinized, and reseeded into 6-well plates at 500–1000 cells per well. The colonies were allowed to form for 10–15 days. At the end of the culture, cells were washed with PBS, fixed with methanol for 30 min, then stained with 0.5% crystal violet overnight. After careful washing, the images were taken.

#### 3.2.7. In Vivo Assay

All animal studies were conducted under the approval of the Experimental Animal Management Committee of Nankai University (No. NKUEC20190228). Six- to eight-week-old female BALB/c nude mice were purchased from Beijing HFK Bioscience Company. Mice were fed in an SPF-level laboratory animal room. The mice were kept in independent ventilation cages with autoclaved corncobs at the bottom of the cage as bedding. The dimensions of the cages were 390 × 180 × 180 mm. Environmental factors: Average air temperature ranged from 20 to 26 °C with average relative air humidity ranging from 40 to 70%. Noise was below 60 decibels. Ventilation times were more than 15 times/hour. Air velocity was 0.1~0.2m/s. The ammonia concentration was below 14/(mg/m³). Working illumination was 150~300LX and animal illumination was 15~20LX. The nutritional components of mouse food include: water content: ≤10%, crude protein: ≥18%, crude fat: ≥4%, crude fiber: ≥5%, minerals: ≥4%, calcium: 1.0–1.8%, Phosphorus: 0.6-1.2%. The water that mice drink is sterilized by the animal drinking system.

Cells SiHa were harvested during the exponential-growth phase, washed 3 times with serum-free medium, followed by resuspension at a concentration of 2 × 10^6^ per mL. A total of 100 μL of cell suspension was injected into SCID mice subcutaneously. After the tumors had grown to 100−150 mm^3^, all the mice were randomized into 5 groups (5 mice for each group, 25 mice total). The mice were treated daily via oral gavage administration with **12O** (5, 10, or 20 mg kg^−1^ d^−1^), cisplatin (20 mg kg^−1^ d^−1^), or vehicle. The dosage adopted in our experiment concurred with the previous literature [21,25] and showed prominent antitumor efficacy. Mice were monitored for side effects every day. Body weights and tumor sizes were determined every other day and volume was also observed. At the end-point of the study, all mice had survived. Mice were euthanized and the subcutaneous tumor tissues were stripped and collected. Tumor measurements were calculated using a digital vernier caliper, and the volumes were determined using the following calculation: (short^2^) × long × 0.5. Inhibition rate of tumor growth was calculated using the following formula: 100 × {1 − [(tumor volume final-tumor volume initial) for **12O**-treated group]/ [(tumor volume final-tumor volume initial) for the vehicle-treated group]} [23].

#### 3.2.8. Hematoxylin and Eosin (H&E)

H&E staining was performed on the formalin-fixed, paraffin-embedded orthotopic mice tumor tissues. Tumor tissue sections were deparaffinized, counterstained with hematoxylin and eosin, then observed under a light microscopy (Olympus).

#### 3.2.9. Statistical Analysis

Statistical analysis results were analyzed values by GraphPad Prism software (GradPad Systems Inc., San Diego, CA). For Student *t* test and ANOVA *p* < 0.05 was considered statistically significant. Values were expressed as means ± SEM. Significance was determined by *χ^2^* test, others were determined by Student’s t-test. A value of *p* < 0.05 was used as the criterion for statistical significance. *** indicates significant difference with *p* < 0.001, ** indicates *p* < 0.01, * indicates *p* < 0.05.

## 4. Conclusions

In summary, we have described the discovery of **12O** as potent a novel multi-target kinase inhibitor with IC_50_ value in the nanomolar range against solid tumor. Docking evaluations provided insight into enzyme inhibitory interactions of **12O** and explained that the introduction of triazole moiety was a key strategic group in the improvement of CDK inhibitory activities and antiproliferative activities. Compound **12O** displayed an outstanding inhibition against several kinases considered important regulators of cancer progression. In cellular assays, **12O** exhibited excellent cytotoxicity and anti-proliferative activity on various cancer cell lines. In addition, **12O** arrested the cell cycle in the G2 phase. Furthermore, the anticancer activity of **12O** was also associated with inducing apoptosis. In in vivo pharmacology evaluations of compound **12O** showed significant antitumor activity (TGI = 79.29%) in an SiHa xenograft model at the dose of 20 mg/kg. These findings implied that molecule **12O** deserves further research.

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
