# Peer review of "Discovery of 12O—A Novel Oral Multi-Kinase Inhibitor for the Treatment of Solid Tumor"

_molecules, 2020, doi:10.3390/molecules25215199_

Round 1

Reviewer 1 Report

In this research paper, the authors have investigated the anti-proliferative effect of pyrimidine-benzotriazole derivatives using solid tumor cells, cervical SiHA) and ovarian cancer cell lines (SKOV3). Initial investigation was performed using CCK-8 assay.  Amon the derivatives tested, 12O was most effective in inhibiting cancer cell proliferation and this was confirmed using multiple cancer cell lines representing cervical, breast, ovarian and lung cancer cells.  Later, the authors investigated the ability of 12O to inhibit various kinases such as CDK4, KDR and PDGFR-related kinases.    Their result show that, 12O potently inhibitedCDK2, CDK5 CDK9 and FLT3.  Significant inhibition was also observed against KDR family members such as VEGFR2, VEGFR1 and VEGFR3, although the IC50 of the inhibition varied slightly among these kinases.  Through molecular docking, the authors also identified key amino acids in the kinases that interact with the 12O.  In subsequent experiments, the anti-proliferative effects of 12O was confirmed using colony formation assay, and through flow cytometry analysis, they showed that 120 induces cell cycle arrest at G2/M.  They also investigated the anti-tumor efficacy of 120 in in vivo models using SiHA xenograft mice.  They demonstrated the tumor growth inhibitions of 51.25%, 65.52%, and 79.29% at doses of 5, 10, and 20 mg/kg respectively. The authors suggest that 12) molecule has the potential in the treatment of cancer.        

Minor comments:

  1.  The authors have performed colony formation assay in SiHa cells.  It will be good to see if this result is reproducible in another cell line such as SKOV3 cells to confirm the findings.
  2. The methods sections, from line 633 to 643 describes Molecular docking studies, Kinase inhibitor assays, and cytotoxicity assays in few sentences.  The methods written is too brief, and please give details of the experimental protocols in detail.
  3. Under colony formation assay (line 657), the authors state that 300 cells were seeded in 6 well plate and incubation for 5 days followed by fixing cells and staining.  Five days of incubation may result in very few cells being present in each colony and the size of the colony may not be big enough to stain and observe visually?  In general, for colony formation assays, the plates are incubated 2 to 3 weeks before staining.   Please explain incubation for 5 days is correct, or is it 15 days?  It could be typo error.    

12O compound appears to exert an inhibitory effect over a several kinases.  Although its efficacy in the inhibition of tumor growth has been demonstrated in animal models, it is likely that administration of the compound to humans may cause serious side effects. This possibility should be discussed in the discussion section. 

Author Response

Point 1:  The authors have performed colony formation assay in SiHa cells.  It will be good to see if this result is reproducible in another cell line such as SKOV3 cells to confirm the findings.

Response 1: According to reviewer’s suggestion, we performed colony formation assay in SKOV3 cells as below. These results confirmed the findings that 12O groups significantly reduced clonogenicity compared with the other treatments in two cell lines.

The results have been added in Figure S3. We have added some discussion in the revised manuscript as below: (lines 185-187).

“We also performed colony formation assay in SKOV3 cells at concentrations according to IC 50 values to confirm the result (Figure S3).”

Point 2:  The methods sections, from line 633 to 643 describes Molecular docking studies, Kinase inhibitor assays, and cytotoxicity assays in few sentences.  The methods written is too brief, and please give details of the experimental protocols in detail.

Response 2:  We have added more details of the experimental protocols in the revised manuscript as below: (lines 642-688).

“3.2.1 Molecular Docking

All the calculations were carried out using the platform of Discovery Studio 3.1 (DS 3.1, Accelrys Inc., San Diego, CA). The receptor protein was prepared by the DS 3.1 software package. Docking studies were finished by Discovery Studio 3.1 to explore the predicted binding modes of compound 12O in CDK2 (PDB entry: 1H1P), CDK5 (PDB entry: 1UNG), CDK9 (PDB entry: 4BCF), FLT3 (PDB entry: 5X02), respectively. Hydrogen atoms were added to proteins using Discovery Studio 3.1. The images were created by PyMOL.

3.2.2 Kinase Inhibition Assays

Kinase inhibition profiles were determined using KinaseProfiler services provided by Eurofins, and ATP concentrations used are the Km of corresponding kinases.

CDK1/cyclinB (h) [CDK2/cyclinA (h), CDK3/cyclinE (h), CDK4/CyclinD(h), CDK5/p25 (h), CDK6/cyclinD3 (h)] is incubated with 8 mM MOPS pH 7.0, 0.2 mM EDTA, 0.1 mg/mL histone H1, 10 mM MgAcetate and [gamma-33P]-ATP (specific activity and concentration as required).

CDK7/cyclinH/MAT1 (h) is incubated with 8 mM MOPS pH 7.0, 0.2 mM EDTA, 500 uM peptide, 10 mM MgAcetate and [gamma-33P]-ATP (specific activity and concentration as required).

CDK9/cyclinT1 (h) is incubated with 8 mM MOPS pH 7.0, 0.2 mM EDTA, 100 uM KTFCGTPEYLAPEVRREPRILSEEEQEMFRDFDYIADWC, 10 mM MgAcetate and [gamma-33P]-ATP (specific activity and concentration as required).

Flt1 (h) is incubated with 8 mM MOPS pH 7.0, 0.2 mM EDTA, 250 uM KKKSPGEYVNIEFG, 10 mM MgAcetate and [g- 33P]-ATP; Flt3 (h) [Abl (h)] is incubated with 8 mM MOPS pH 7.0, 0.2 mM EDTA, 50 uM EAIYAAPFAKKK, 10 mM MgAcetate and [gamma-33P]- ATP ; Flt4 (h) is incubated with 8 mM MOPS pH 7.0, 0.2 mM EDTA, 500 uM GGEEEEYFELVKKKK, 10 mM MgAcetate and [gamma-33P]-ATP.

c-Kit (h) is incubated with 8 mM MOPS pH 7.0, 0.2 mM EDTA, 10 mM MnCl2, 0.1 mg/mL poly(Glu, Tyr) 4:1, 10 mM MgAcetate and [gamma-33P]-ATP.

FGFR1 (h) is incubated with 8 mM MOPS pH 7.0, 0.2 mM EDTA, 250 uM KKKSPGEYVNIEFG, 10 mM MgAcetate and [gamma-33P]-ATP.

Src (1-530) (h) is incubated with 8 mM MOPS pH 7.0, 0.2 mM EDTA, 500 uM GGEEEEYFELVKKKK, 10 mM MgAcetate and [gamma-33P]-ATP.

KDR (h) is incubated with 8 mM MOPS pH 7.0, 0.2 mM EDTA, 0.33 mg/mL myelin basic protein, 10 mM MgAcetate and [gamma-33P]-ATP.

PDGFR-alfa(h) [PDGFR-beta(h)] is incubated with 8 mM MOPS pH 7.0, 0.2 mM EDTA, 0.1 mg/mL poly(Glu, Tyr) 4:1, 10 mM MnCl2, 10 mM MgAcetate and [gamma-33P]-ATP.

The reactions of all above are initiated by the addition of the Mg/ATP mix. After incubation for 40 minutes at room temperature, the reaction is stopped by the addition of phosphoric acid to a concentration of 0.5%. 10 µL of the reaction is then spotted onto a P30 filtermat and washed four times for 4 minutes in 0.425% phosphoric acid and once in methanol prior to drying and scintillation counting.

3.2.3 Cytotoxicity Assay

The viability of cells as determined using the CCK8 assay method. Cells were seeded (2000-3000 cells per well) in 96-well plates. After incubation for 24h in serum-containing media, the cells were treated with drugs at different concentrations diluted with culture medium for 72h at 37℃ with a 5% CO2 atmosphere. Then, 10μL CCK8 regent (Promega, WI) were added to each well, and the plates were incubated for 1-4h at 37℃. Finally, absorbance values of test wells (AS), control wells (AC) and blank wells (Ab) at 450nm were read using the Microplate reader (Promega, WI). Inhibition ratio were calculated as followed: [(AC - AS)/ (AC - Ab)] X 100%. IC50 values were calculated using GraphPad software [17]”

Point 3:  Under colony formation assay (line 657), the authors state that 300 cells were seeded in 6 well plate and incubation for 5 days followed by fixing cells and staining.  Five days of incubation may result in very few cells being present in each colony and the size of the colony may not be big enough to stain and observe visually?  In general, for colony formation assays, the plates are incubated 2 to 3 weeks before staining.  Please explain incubation for 5 days is correct, or is it 15 days?  It could be typo error.    

Response 3: Sorry for our negligence. We revised the experimental protocol of colony formation assay in the revised manuscript (lines 702-706) as below:

“3.2.6 Colony Formation Assay

Cells treated with indicated compounds or vehicle were washed with PBS, trypsinized, and reseeded into 6-well plates at 500-1000 cells per well. The colonies were allowed to form for 10-15 days. At the end of the culture, cells were washed with PBS, fixed with methanol for 30 min, then stained with 0.5% crystal violet overnight. After careful washing, the images were taken.”

Point 4: 12O compound appears to exert an inhibitory effect over a several kinases.  Although its efficacy in the inhibition of tumor growth has been demonstrated in animal models, it is likely that administration of the compound to humans may cause serious side effects. This possibility should be discussed in the discussion section. 

Response 4: We have added more discussion in the revised manuscript (lines 228-234) as below:

12O as a novel multi-target kinase inhibitor effectively inhibited tumor growth of mice without obvious toxicity. Additional kinase activities of 12O intimately associated with the growth, survival, and metastasis in tumor cells, may contribute to the antitumor activity, but may also cause side effects. Multi-target kinase inhibitor extensively used in clinic cancer therapy, while hampered by associated adverse reactions and side-effect. Thus, further evidence is needed. An in-depth study of compound 12O is ongoing in our laboratory and will be reported in due course.”

Special thanks to you for your good comments!

Reviewer 2 Report

The authors in this article present clear evidence that a pyrimidine-benzotriazole derivative, more specifically a 12O molecule, has antiproliferative activities in the SiHa cell lines. These activities occurred both in an in vitro assay, where it demonstrated a decrease in cell proliferation, colony formation and cell cycle progression, as well as in the in vivo assay, where there was a clear evidence of decreased tumor growth. Studies show to be well thought out and carefully placed, however, there are some improvements necessary for a replicability of the results obtained, more specifically:

Regarding the oral administration of 12O, it was not clear how it was done correctly, that is, if there was an incorporation in the food or water that the animals ingest or what procedure was used. Still on this topic it suggested that the number of times this administration was necessary for the animals needed to be complete in the material and methods and not in the results and discussion.

In any animal protocol, there are some aspects that must be broken down and clarified for a better veracity of the results, including parameters such as temperature, humidity, light intensity, dimensions of the cages and humidity conditions. There is also no reference to food used and the diet that the animals were subjected to. The authors make no reference to the method of euthanasia and how they collected tumors from animals. The authors’ state that they were approved Experimental Animal Management Committee of Nankai University, however do not make reference to the approval number. Authors should also indicate how many animals were used and if all animals survive. Authors should explain why were used these doses of 12O, and not other,

It seems to me that the topic related to the description of the synthesis of the compound is too long and not very interesting for the readers of this magazine.

Authors should add a bibliographic reference to the formula used to measure tumors.

In parameter 3.2.9 in the statistical analysis the reference to the city and country of the GraphPad Prism program is missing. Finally, I suggest the change of nuclei mainly in graph A of Figure 5 in order to make the different groups noticeable.

Concerning the results, the figure 5 e) is inconclusive.

Author Response

Point 1: Regarding the oral administration of 12O, it was not clear how it was done correctly, that is, if there was an incorporation in the food or water that the animals ingest or what procedure was used. Still on this topic it suggested that the number of times this administration was necessary for the animals needed to be complete in the material and methods and not in the results and discussion.

Response 1:  We have added detail description in the material and methods in the revised manuscript (lines 707-732) as below:

“All animal studies were conducted under the approval of the Experimental Animal Management Committee of Nankai University (No. NKUEC20190228). 6- to 8-week-old female BALB/c nude mice were purchased from Beijing HFK Bioscience Company. Mice were feeded in SPF-level laboratory animal room. The mice are kept in independent ventilation cages with autoclaved corncobs at the bottom of the cage as bedding. The dimensions of the cages are 390 x 180 x 180 mm. Environmental factors: Average air temperature ranged from 20 °C to 26 °C with average relative air humidity ranging from 40 to 70%. Noise below 60 decibels. Ventilation times are more than 15 times/hour. Air velocity are 0.1~0.2m/s. The ammonia concentration is below 14/(mg/m³). Working illumination is 150~300LX and animal illumination is 15~20LX. The nutritional components of mouse food include: water content: ≤10%, crude protein: ≥18%, crude fat: ≥4%, crude fiber: ≥5%, minerals: ≥4%, calcium: 1.0-1.8%, Phosphorus: 0.6-1.2%. The water that mice drink is sterilized by the animal drinking system.

 Cells SiHa were harvested during the exponential-growth phase, washed 3 times with serum-free medium, followed by resuspension at a concentration of 2 × 106 per mL. A total of 100 μL of cell suspension was injected into SCID mice subcutaneously. After the tumors had grown to 100−150 mm3, all the mice were randomized into 5 groups (5 mice for each group, 25 mice total). The mice were treated daily via oral gavage administration with 12O (5, 10, or 20 mg kg−1 d−1), cisplatin (20 mg kg−1 d−1), or vehicle. The dosage adopted in our experiment referenced to the previous literature [15, 28] and showed prominent antitumour efficacy. Mice were monitored for side effects every day. Body weights and tumor size were determined every other day. volume was also observed. At the end-point of the study, all mice survive. Mice were euthanized and the subcutaneous tumour tissues were stripped and collected. Tumor measurements were used using a digital vernier caliper, and the volumes were determined using the following calculation: (short2) × long × 0.5. Inhibition rate of tumor growth was calculated using the following formula: 100 × {1 - [(tumor volume final-tumor volume initial) for 12O-treated group]/ [(tumor volume final-tumor volume initial) for the vehicle-treated group]}[17].”

Point 2: In any animal protocol, there are some aspects that must be broken down and clarified for a better veracity of the results, including parameters such as temperature, humidity, light intensity, dimensions of the cages and humidity conditions. There is also no reference to food used and the diet that the animals were subjected to. The authors make no reference to the method of euthanasia and how they collected tumors from animals. The authors’ state that they were approved Experimental Animal Management Committee of Nankai University, however do not make reference to the approval number. Authors should also indicate how many animals were used and if all animals survive. Authors should explain why were used these doses of 12O, and not other,

Response 2:  According to the Reviewer’s suggestion, we have added detail description of the animal protocol in the revised manuscript as below.

The dosage adopted in our experiment referenced to the previous literature: As reported by KC Goh, et al (Leukemia, 2012, 26, 236–243), there was also a multi-kinase inhibitor TG02 that inhibited CDKs together with JAK2 and FLT3. In the combined 18-tumor panel, the mean IC50 for TG02 was 0.19 μM. 10, 20 and 40 mg/kg was used for TG02 in AML mice model; As reported by our group previously (European Journal of Medicinal Chemistry,2019, 181, 111541), there was also a multi-kinase inhibitor 5J. 5J had significant cancer cell inhibitory activity with IC50 around 0.050 μM. 10, 20 and 40mg/kg was used for 5J in mice model. Thus, as compound 12O in this paper showed remarkable inhibitory effect against in cervical cancer cell line (SiHa) with IC50 value as 0.009 µM, we performed 5,10, 20 mg/kg in the in vivo assay and showed prominent antitumour efficacy. We have added some explanation in the revised manuscript (lines 707-732) as below.

“3.2.7 In Vivo Assay

All animal studies were conducted under the approval of the Experimental Animal Management Committee of Nankai University (No. NKUEC20190228). 6- to 8-week-old female BALB/c nude mice were purchased from Beijing HFK Bioscience Company. Mice were feeded in SPF-level laboratory animal room. The mice are kept in independent ventilation cages with autoclaved corncobs at the bottom of the cage as bedding. The dimensions of the cages are 390 x 180 x 180 mm. Environmental factors: Average air temperature ranged from 20 °C to 26 °C with average relative air humidity ranging from 40 to 70%. Noise below 60 decibels. Ventilation times are more than 15 times/hour. Air velocity are 0.1~0.2m/s. The ammonia concentration is below 14/(mg/m³). Working illumination is 150~300LX and animal illumination is 15~20LX. The nutritional components of mouse food include: water content: ≤10%, crude protein: ≥18%, crude fat: ≥4%, crude fiber: ≥5%, minerals: ≥4%, calcium: 1.0-1.8%, Phosphorus: 0.6-1.2%. The water that mice drink is sterilized by the animal drinking system.

 Cells SiHa were harvested during the exponential-growth phase, washed 3 times with serum-free medium, followed by resuspension at a concentration of 2 × 106 per mL. A total of 100 μL of cell suspension was injected into SCID mice subcutaneously. After the tumors had grown to 100−150 mm3, all the mice were randomized into 5 groups (5 mice for each group, 25 mice total). The mice were treated daily via oral gavage administration with 12O (5, 10, or 20 mg kg−1 d−1), cisplatin (20 mg kg−1 d−1), or vehicle. The dosage adopted in our experiment referenced to the previous literature [15, 28] and showed prominent antitumour efficacy. Mice were monitored for side effects every day. Body weights and tumor size were determined every other day. volume was also observed. At the end-point of the study, all mice survive. Mice were euthanized and the subcutaneous tumour tissues were stripped and collected. Tumor measurements were used using a digital vernier caliper, and the volumes were determined using the following calculation: (short2) × long × 0.5. Inhibition rate of tumor growth was calculated using the following formula: 100 × {1 - [(tumor volume final-tumor volume initial) for 12O-treated group]/ [(tumor volume final-tumor volume initial) for the vehicle-treated group]}[17].”

Point 3: It seems to me that the topic related to the description of the synthesis of the compound is too long and not very interesting for the readers of this magazine.

Response 3:  We have revised the description of the synthesis of the compound in the revised manuscript as below: (lines 88-95).

“2.1. Chemistry

The synthetic routes for all novel compounds were shown in Scheme 1 and 2. As shown in Scheme1, commercial compound 4-bromo-2-fluoro-1-nitrobenzene (1) reacted with isopropylamine to give intermediate 2, which underwent reduction reaction to provide compound 3. 3 was converted to 4 by closing the ring in the presence of conc.HCl and aq NaNO2. Then, 4 was heated with bis(pinacolato)diboronin to give 5, which yield 6. As shown in Scheme 2, commercially available 7 reacted with aniline or substituted aniline to afford 8A-8P, which afforded 10A-P. Then final compounds 12A-P were generated by a palladium catalyzed cross-coupling reaction of compounds 10A-P with compound 6.”

Point 4: Authors should add a bibliographic reference to the formula used to measure tumors.

Response 4: We have added a bibliographic reference into the Material and Method part in the revised manuscript (line 732).

Point 5:  In parameter 3.2.9 in the statistical analysis the reference to the city and country of the GraphPad Prism program is missing. Finally, I suggest the change of nuclei mainly in graph A of Figure 5 in order to make the different groups noticeable.

Response 5:  Sorry for our negligence. We have added the city and country of the GraphPad Prism program into the Material and Method part in the revised manuscript as below: (lines 738-739). In order to make the different groups noticeable, we adjusted the color of the curves Figure 5a.

3.2.9 Statistical Analysis

Statistical analysis results were analyzed values by GraphPad Prism software (GradPad Systems Inc., San Diego, CA). For Student t test and ANOVA P< 0.05 was considered statistically significant. Values were expressed as means ± SEM. Significance was determined by χ2 test, others were determined by Student’s t-test. A value of P < 0.05 was used as the criterion for statistical significance. ***indicates significant difference with P < 0.001, ** indicates P < 0.01, * indicates P< 0.05.

Figure 5a

Point 6: Concerning the results, the figure 5 e) is inconclusive.

Response 6:  We have added some description for Figure 5e in the revised manuscript as below: (lines 224-226).

“It was showed that the nuclei of tumor cells in vehicle controls were large and hyperchromatic, while the nuclei of 12O-treated tumor cells were pyknotic. HE staining results in tumor tissues treated with 12O further demonstrated the inhibition of tumor growth.”

Special thanks to you for your good comments!

Round 2

Reviewer 1 Report

In this research paper, the authors have investigated the anti-proliferative effect of pyrimidine-benzotriazole derivatives using solid tumor cells, cervical SiHA) and ovarian cancer cell lines (SKOV3). Initial investigation was performed using CCK-8 assay.  Amon the derivatives tested, 12O was most effective in inhibiting cancer cell proliferation and this was confirmed using multiple cancer cell lines representing cervical, breast, ovarian and lung cancer cells.  Later, the authors investigated the ability of 12O to inhibit various kinases such as CDK4, KDR and PDGFR-related kinases.    Their result show that, 12O potently inhibitedCDK2, CDK5 CDK9 and FLT3.  Significant inhibition was also observed against KDR family members such as VEGFR2, VEGFR1 and VEGFR3, although the IC50 of the inhibition varied slightly among these kinases.  Through molecular docking, the authors also identified key amino acids in the kinases that interact with the 12O.  In subsequent experiments, the anti-proliferative effects of 12O was confirmed using colony formation assay, and through flow cytometry analysis, they showed that 120 induces cell cycle arrest at G2/M.  They also investigated the anti-tumor efficacy of 120 in in vivo models using SiHA xenograft mice.  They demonstrated the tumor growth inhibitions of 51.25%, 65.52%, and 79.29% at doses of 5, 10, and 20 mg/kg respectively. The authors suggest that 12) molecule has the potential in the treatment of cancer.        

Author Response

Dear reviewer:
We have checked the whole manuscript and some spell checks were corrected
Thank you very much for your time in advance.
Sincerely yours,
Yan Fan

Reviewer 2 Report

Authors performed the requested corrections. 

I suggest to put in all images from figure 5 the Scale bar.

The manuscript can now be accepted for publication.

Author Response

Dear Editor:

Manuscript ID: molecules-945270

Thank you for the review. After carefully considering the comments of the reviewer, we have revised the original manuscript.

We appreciate so much that you gave us a chance of minor revision to improve our manuscript to a level suitable for publication in your journal.

Reviewer 2  

Point 1:  Authors performed the requested corrections.

I suggest to put in all images from figure 5 the Scale bar.

The manuscript can now be accepted for publication. 

Response 1: According to reviewer’s suggestion, we put the Scale bar in the images of figure 5 as below. We have added the detailed size of the Scale bar in the legend in the revised manuscript. 

Thank you very much for your time in advance.

Sincerely yours,

Yan Fan
